



# Similar importance of inter-tree and intra-tree variations in wood density observations in Central Europe

Hui Yang[1], Krzysztof Stereńczak[2,3], Zbigniew Karaszewski[4], Nuno Carvalhais[1]

[1] Department of Biogeochemical Integration, Max Planck Institute for Biogeochemistry, 07745 Jena, Germany
[2] Instytut Badawczy Leśnictwa, ul. Braci Leśnej 3, Sękocin Stary, 05-090 Raszyn
[3] Forest Research Institute, Braci Leśnej 3 Street, Sękocin Stary, 05-090 Raszyn
[4] Department of Wood Investigation and Application, Łukasiewicz Research Network-Wood Technology Institute,
Winiarska 1, 60-654 Pozna´n, Poland

*Correspondence to*: Hui Yang (huiyang@bgc-jena.mpg.de)

**Abstract.** Wood density is a crucial variable linked to mechanical, physiological, and ecological properties. In this study, we analyzed an extensive dataset of over 48,000 wood density samples collected from 2,920 trees. Our aim was to explore variations in wood density, at both inter-tree and intra-tree levels, along with the factors contributing to these variations. Inter-tree variations reveal significant differences in wood density among eight dominant species, highlighting their role in shaping wood density. As tree species exhibit specific spatial distributions associated with microhabitats, we anticipated a link between
wood density distribution and microhabitat. Using a feature selection approach and random forest model, we identified six predictors, including satellite-based vegetation indexes, topographic variables, and soil sand content, capable of predicting 91% of spatial wood density variations. The Normalized Difference Vegetation Index (NDVI) positively representing the amount of carbon within trees correlated with wood density, while the Normalized Difference Water Index (NDWI), reflecting water content, and soil sand content showed negative associations. Geomorphons and soil sand context provided insights into
wood density variations and specific landforms. Lower wood density values were linked to landforms with low geomorphons (summit, ridge, or shoulder), whereas higher wood density was found in landforms with high geomorphons (valley, depression, or hollow areas). Furthermore, our study highlighted the importance of considering intra-tree variation, a facet often overlooked in previous research. Interestingly, the magnitude of intra-tree variation is comparable to, and in some species even exceeds, that of inter-tree variations. The intra-tree wood density samples display significant differences both vertically along
the height and radially from the center to the bark zones of trees. These variations are influenced by tree growing strategy, living conditions, and physiological structure. In summary, our research delved into the multifaceted features of wood density, shedding light on critical aspects of this fundamental variable.



## 1 Introduction

Wood density, the ratio of the oven-dry mass of wood sample to its green volume, is a fundamental trait describe the carbon apportioning within the trees. It serves as a key indicator for various ecological and physiological processes, such as tree growth and mortality rates and hydraulic properties (Chave et al. 2009). Firstly, wood density is an indispensable input for estimating above-ground biomass or carbon stocks based on the allometric equations, whether at the large-scale utilizing satellite earth observation data (e.g., ESA GlobBiomass product; Santoro et al. 2022) or at the small- or regional-scale using

forest inventory data (e.g., forest resource assessment report from the Food and Agriculture Organization; FAO 2020). Therefore, assessing wood density has the potential to have a positive impact on the accuracy and precision of forest biomass stock assessments and associated national greenhouse gas inventories. Moreover, wood density exhibits a relationship with tree mortality rate and the carbon turnover within ecosystem (King et al. 2006; Kraft et al. 2010), for example, Chao et al. (2008) found contrasting tree mortality rates in eastern and western Amazon forests, attributed to variations in wood density,

with lower density associated with higher mortality rates. In particular, recent studies indicate that wood density is associated with the resistance and resilience of forests to natural and anthropogenic disturbances such as drought and fires (Anderegg et al. 2016; Brando et al. 2012; Liang et al. 2020). This highlights the importance of understanding wood density to predict vegetation and carbon cycle dynamics under changing climate.

Large variability in wood density between trees has been reported. Firstly, wood density varies considerably across different tree species, genera, or families. Thurner et al. (2014) assessed the wood density measurements from Global Wood Density Database (Chave et al. 2006; Zanne et al. 2009), and found that, on average, broadleaf trees have higher wood density than needleleaf trees, but even within the same genus, significant divergence in wood density can be observed. Furthermore, the variation in wood density is closely linked to tree growth conditions, which encompass factors such as climate, nutrient

availability, and soil characteristics. It is important to note that these influencing factors can vary from one region to another. For example, previous regional studies have reported that wood density tends to increase with higher growth temperature (Thomas et al. 2005; Sweson and Enquist 2007) and lower elevation (Sungpalee et al. 2009), and the role of soil water availability differs between wet and dry biomes, where it has a negative impact on wood density in wet biomes but a positive impact in dry biomes (Rocha et al. 2020). Moreover, at the microscopic level, wood density is influenced by the characteristics

of individual tracheid cells. Thin cell walls, resulting from fast growth and lower competition with neighboring trees, are typically associated with low wood density. Conversely, thick cell walls, a consequence of slower growth, are related to high wood density (Gyrc et al. 2011). This suggests that the stage of tree succession and/or the growth strategy during different developmental stages also play a role in shaping wood density gradients.



In this study, we use a novel dataset comprising wood density measurements collected from forests in Poland to investigate both inter-tree and intra-tree variations in wood density. The primary objectives of this study are as follows:

1. To determine the magnitude of inter-tree variations in wood density. We aim to explore how factors such as leaf type, tree family, tree species, and location contribute to the observed inter-tree variations. Additionally, we seek to understand how biotic and abiotic factors influence wood density variations between trees.

2. To examine how wood density changes with tree height (vertical density profiles), radius (radial density profiles), and different directions (northern or southern discs) within individual trees. We aim to explore the underlying reasons behind these different vertical and radial density profiles within individual trees.

3. To compare the extent of inter-tree and intra-tree variations in wood density for the analyzed tree species or forest plots. We aim to determine which variation is larger and provide recommendations for estimating wood density at a

large-scale.

In many previous works, wood density formulas were derived from a limited data set that included, for example, only older trees of a particular species that had grown under certain conditions. As a result, it was difficult to understand the relationships between many environmental variables and variation in wood density. The dataset used in this work is unique for Central Europe and, although it was only collected for Poland, covers age, habitat and height distributions characteristic of this part of

the world. By addressing these research objectives, we aim to enhance our understanding of wood density variations both between and within trees, and provide insights into the estimation of wood density on a broader scale.

## 2 Methods

### 2.1 Study site and wood density sample collection

Our dataset comprises more than 48,000 wood density samples that were measured in the year 2018 for 2,920 trees from 391

forest plots in Poland (Figure 1). The sampled trees, aged over 5 years, encompass a range of species and their relevant information such as latitude, longitude, age, and species type were recorded. The dataset consists of eight common tree species (belonging to three families): *Pinus sylvestris* (Pinaceae), *Picea abies* (Pinaceae), *Abies alba* (Pinaceae), *Larix decidua* (Pinaceae), *Quercus robur* (Fagaceae), *Fagus sylvatica* (Fagaceae), *Betula pendula* (Betulaceae), and *Alnus glutinosa* (Betulaceae). Moreover, specific divisions within the dataset exist for certain species. For example, the plots of *Pinus sylvestris*

and *Quercus robur* are categorized into two groups based on soil fertility, denoted as "low fertile soils" and "fertile soils" respectively. Similarly, the plots of *Picea abies* and *Fagus sylvatica* are classified into two groups according to the elevation of the plots, labeled as "lowlands" and "highlands and mountains" respectively. These distinctions within the species-specific plots add further complexity to the dataset and enable more detailed analysis.

Based on the leaf type and leaf habit, the eight species in our analysis can be classified into three plant function types (PFTs) categories: evergreen needleleaf forest (ENF), deciduous needleleaf forest (DNF) and deciduous broadleaf forest (DBF). One





family, Pinaceae, has both evergreen needleleaf and deciduous needleleaf species. Taking into account both the families and PFTs, the eight species can be further divided into four types, i.e. Pinaceae_ENF, Pinaceae_DNF, Fagaceae_DBF and Betulaceae_DBF. Regarding the ages of the trees, all trees can be classified into nine age classes: The age of trees are 0-20

years, 20-40 years, 40-60 years, …, 140-160 and 160-180 years, respectively. Note that the age of the tree is determined by counting the rings on wood discs obtained from the bottom of a trunk. Furthermore, the trees can be classified into seven height classes and six diameter-at-breast-height (DBH) classes: The heights classes are defined as <10 meters, 10-15 meters, 15-20 meters, 20-25 meters, 25-30 meters, 30-35 meters and >35 meters. The DBH of trees are defined as <100 centimeters, 100-200 centimeters, 200-300 centimeters, 300-400 centimeters, 400-500 centimeters, and >500 centimeters.

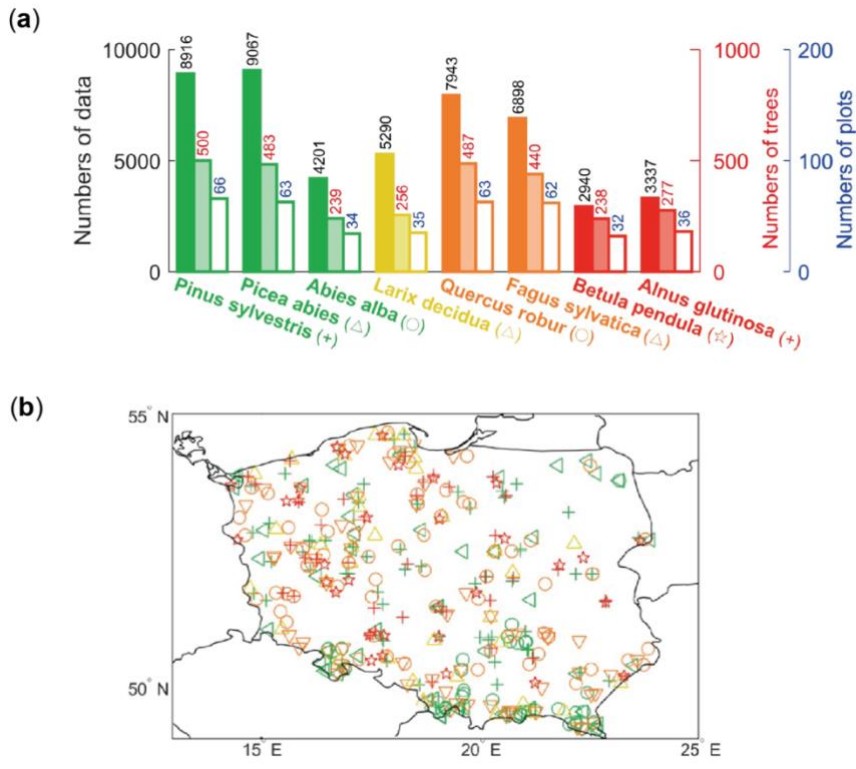

**Figure 1: (a) The number of wood density samples (dark-colored bars), trees (light-colored bars), and forest plots (transparent bars) for each of eight species. (b) The geographical location of 391 forest plots in Poland (See panel (a) for details on notations)..**

For the analysis of the intra-tree variation in wood density, a total of 1,886 trees were included, and for each tree, more than 30 wood samples were taken using a sharp increment borer. These trees were specifically selected as they were dead but had

not undergone wood drying. Each tree was divided into three equal parts, and a disc was obtained from the middle of each part. The three discs were labeled as "bottom," "middle," and "top" based on their respective positions within the tree. Each tree disc was further sampled along two radial directions, namely north and south. The sampling was conducted from the inner



zone to the outer zone of the discs. The number of samples obtained for each disc varied depending on the width of the disc. However, it is worth noting that each disc contained more than 10 samples along these radial directions.


## 2.2 Study site and wood density sample collection

To assess the inter-tree variations, we computed the mean wood density for each tree. Subsequently, we employed analysis of variance (ANOVA) to partition the overall variations in tree-level wood density (n = 2,920) across different levels including leaf habit, leaf type, family, species, and age classes. The total variance is calculated as:

$$\sum_{i=1}^{t} \sum_{j=1}^{n_i} (X_{ij} - \bar{X})^2 , \qquad (1)$$

where $X_{ij}$ is the $j$th wood density from class $i$, and there are $n_i$ wood density samples in class $i$, and $\bar{X}$ is the average of all the wood density samples. And variance explained by leaf habit/leaf type/family/species/age is calculated as:

$$\sum_{i=1}^{t} n_i (\overline{X_{i\cdot}} - \bar{X})^2 , \qquad (2)$$

where $\overline{X_{i\cdot}}$ is the average of wood density of class $i$.


To account for the influence of location, we aggregated tree-level wood density measurements based on geographical proximity. Specifically, we considered two criteria: (1) trees located within a short distance of each other, such as less than 100 meters or 500 meters, and (2) trees falling within the same fine-resolution grid cell, with grid sizes of 0.05° or 0.1°. Using the first criterion, trees were considered to be at the same location if the distance between them was less than 100 meters (or 500 meters). This resulted in the distribution of 2,920 trees across 382 unique locations (or 372 unique locations). Using the second criterion, a pre-defined grid mesh with resolutions of 0.05° or 0.1° was utilized. Trees falling within the same grid cell were considered to be at the same location. To analyze the impact of location, we applied the same ANOVA methodology used previously to partition the total variations in tree-level wood density into different locations.

To investigate the factors that influence the spatial distribution of tree-level wood density, we employed a feature selection method (Jung and Zscheischler 2013) to identify the most significant predictors. Based on this selection, six important covariates were chosen, including vegetation indexes, vegetation water content, soil texture, and topographic characteristics (refer to Table 1). These selected covariates were then used to train a random forest model in the cross-validation analysis. To evaluate the performance of the model, we assessed its efficiency using the Out-of-bag (OOB) $R^2$ metric, which yielded a value of 0.91. Additionally, in order to gain insights into how these selected covariates influence wood density within the random forest model, we computed the SHAP (Shapley Additive exPlanations) values for each covariate. These values represent the difference between the model's prediction and the null model (Lundberg and Lee, 2017). By examining the SHAP values, we can gain a better understanding of the individual contributions of each covariate to the prediction of wood density.





# 3 Results

## 3.1 Inter-tree variation in wood density

For the analysis of the inter-tree variation in wood density, we calculate the tree-level wood density by averaging the wood density samples obtained from 2,920 trees. Figure 2a shows the distribution of tree-level wood density for eight tree species, which are classified into three families, and three PFTs. Overall, the variation in wood density among species is greater than the variation within each species. Consistent with the findings in Thurner et al. (2014), our results indicate that the mean wood density of evergreen needleleaf forests, that is *Pinus sylvestris*, *Picea abies*, *Abies alba* species, is lower compared to deciduous needleleaf forests (*Larix decidua* species), and the mean wood density of needleleaf forests is lower than that of broadleaf forests, including *Quercus robur*, *Fagus sylvatica*, *Betula pendula*, and *Alnus glutinosa* species. When considering tree families, the mean wood density of Pinaceae is slightly lower than that of Betulaceae, and significantly lower than that of Fagaceae. These findings align with the general patterns observed in previous research, highlighting the differences in wood density among tree species and families.

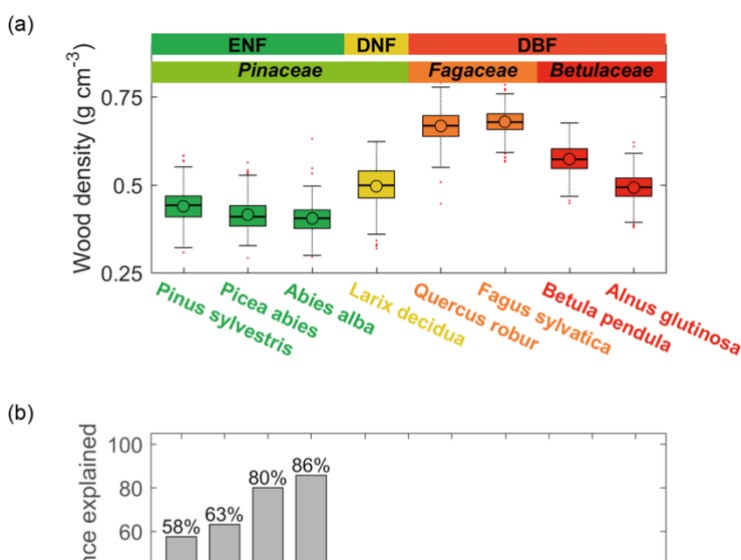

**Figure 2: (a) Boxplots of tree-level wood density for eight species. On each box, the central bar indicates the median and the dot indicates the mean of wood density; the bottom and top edges indicate the 25th and 75th percentiles; the whiskers extend to all data points except outliers (which are plotted individually as small red dots). Eight tree species belong to three families (Pinaceae, Fagaceae and Betulaceae), and can be classified into two categories according to their leaf habit or types: evergreen (E) or deciduous (D) trees, needleleaf (N) or broadleaf (B) trees. Family, leaf habit and leaf type are labelled on the top of boxplot. (b) The fraction of**



**variance of tree-level wood density explained by the leaf habit (two levels), leaf type (two levels), families (three levels), species (eight levels), age (nine levels), height (seven levels), DBH (six levels) and locations. The trees with geographic distance less than 100 or 500 m, or in the same fine-resolution grid cell (i.e., 0.05, 0.1 degree) are considered as in the same location.**

A quantitative analysis indicates that species, families, leaf types, and leaf habits explain a massive portion of the variance in tree-level wood density, accounting for 85%, 80%, 63% and 58% respectively (Figure 2b). In contrast, tree location only explains less than 3% of the variance in tree-level wood density, regardless of the method used to identify trees within the same location (see Methods, section 2.2 Analysis of inter-tree variation in wood density). In other words, the large differences in wood density observed among species tend to diminish when considering the averages within geographical locations. This

can be attributed to the relatively even distribution of the eight primary species (or at least three families) across similar locations in Poland, as shown in Figure 1b.

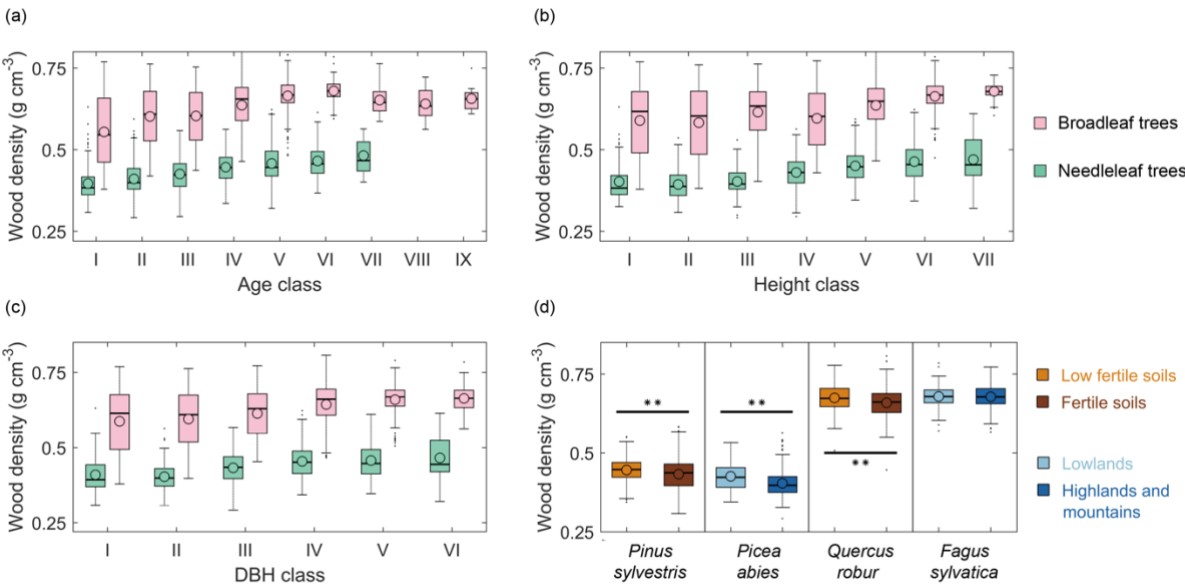

**Figure 3: (a) Boxplots of tree-level wood density for needleleaf and broadleaf trees at nine different age classes. The higher classes, the older ages. (b) Boxplots of tree-level wood density for needleleaf and broadleaf trees with different height classes. The height of tree increases with the number of height class. (c) Boxplots of tree-level wood density for needleleaf and broadleaf trees with different**
**DBH classes. The DBH of tree increases with the number of DBH class. (d) comparison of tree-level wood density for one specific species but growing in the low fertile or fertile soils, or growing at lowlands or highlands (mountains). Two asterisk indicates the significant difference in the mean of two samples (via t-test, 0.001 significance level).**

The contributions of tree age, height, and breast height diameter (DBH) to the variance in wood density between trees are
relatively low, accounting for 6%, 4%, and 2% respectively (Figure 2b). However, when analyzing needleleaf and broadleaf trees separately, their effects become more apparent (Figure 3a-c). First, in broadleaf trees, wood density tends to increase with tree age up to approximately 140 years, after which it stabilizes (Figure 3a). In contrast, needleleaf trees exhibit a continuous increase in wood density with age. Second, both needleleaf and broadleaf trees show an increase in wood density with height and DBH. However, for broadleaf trees, the impact of height is more pronounced in taller trees (height class ≥ 4, i.e., tree





height ≥ 20m). This is because the variance in wood density is greater for trees with lower heights and DBH (Figure 3b-c). In addition, the influence of soil fertility on wood density is noteworthy and consistently observed in two tree species, *Pinus sylvestris* and *Quercus robur*. Trees growing in low-fertility soils exhibit significantly higher wood density compared to those in fertile soils (*t*-test, *p*-val < 0.001; Figure 3d). Unlike soil fertility effects, the effects of elevation or slope on wood density are only evident in needleleaf trees of the *Picea abies* species. Specifically, the wood density of trees in lowland areas is

significantly higher than that in highlands or mountains (*t*-test, *p*-val < 0.001). However, for Fagus sylvatica species, there is no significant difference in wood density between lowland and highland regions (Figure 3d).

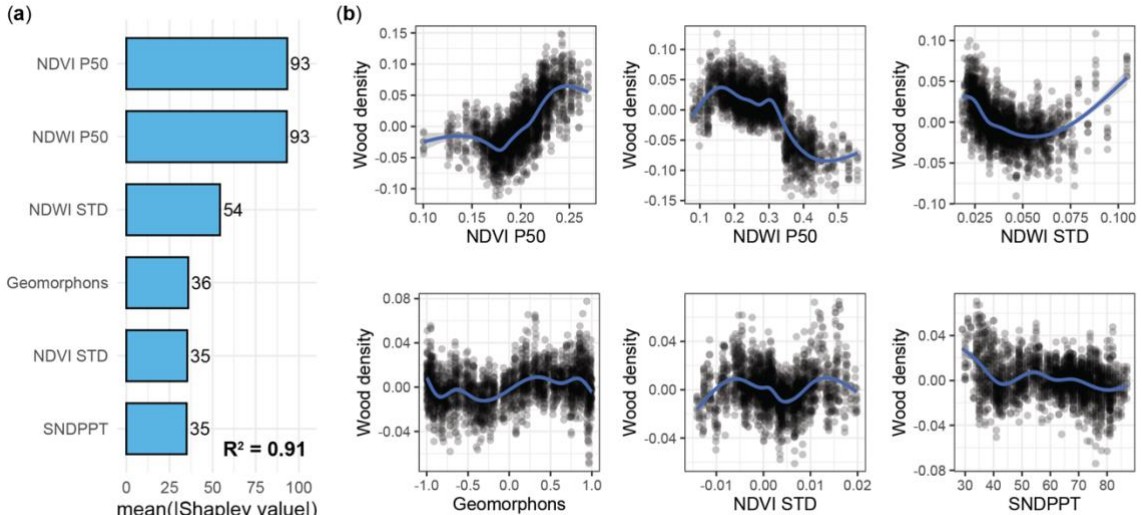

**Figure 4: The factors influencing inter-tree variation in wood density using a random forest (RF) model. (a) Barplot of the mean absolute SHAP values of factors. (b) SHAP dependence plots of the median value (P50) and standard deviation (STD) of the**
**Normalized Difference Vegetation Index (NDVI) and the Normalized Difference Water Index (NDWI), tomographic index of Geomorphons, and soil sand content (SNDPPT).**

Furthermore, we employed a random forest model to predict the observed inter-tree wood density variability and to investigate the importance of influencing factors, including vegetation, soil and topographic properties. Altogether, these factors can

explain 91% of the spatial variations in wood density across trees (Figure 4a) and can distinguish the variability in wood density determined by tree species (Figure S1). Among the predictors, vegetation greenness and water indexes (e.g., NDVI and NDWI) are the most influential factors, followed by topographic geomorphons metric and soil sand content. To further investigate the individual effects of these factors on wood density variation, SHAP dependence plots were generated for the six most influential factors in the random forest model. As shown in Figure 4b, higher NDVI values (indicating greater carbon

content within trees) and lower NDWI values (indicating higher water content within trees) are associated with higher wood density. Geomorphons provide insights into the types of landforms associated with wood density variations. Low geomorphons representing summit, ridge or shoulder correspond to low wood density. Conversely, high wood density is associated with





high geomorphons representing valley, depression, hollow area. Lastly, a negative relationship is observed between wood density and high soil sand content, which reflects lower soil fertility and moisture levels. It is important to note that this finding

contradicts the results obtained for *Pinus sylvestris* and *Quercus robur* in Figure 3d, suggesting species-specific responses to soil fertility.

## 3.2 Intra-tree variation in wood density

*Vertical variations within the trees*

To examine the differences in wood density among the bottom, middle and top parts of trees, we conducted an ANOVA

analysis. The results indicate that about 45% of all the trees exhibit significant vertical variations ($p$-value < 0.1) in wood density (Figure 5a). The percentage of trees with significant vertical variations ($p$-value < 0.1) varied among species, with *Alnus glutinosa* and *Pinus sylvestris* species having the highest percentage (more than 50%), while *Betula pendula* species had the lowest percentage of 15%. Subsequently, we compared the overall distribution of wood density among the three tree parts for those trees with significant vertical variation (see boxplots in Figure 5b). When considering all the samples regardless of

species, there were no significant differences in mean wood density between the bottom, middle, and top parts of the trees, and no clear vertical profiles in wood density could be observed. However, when analyzing individual species, mean wood density was found to vary with tree height. Among the seven species, two types of vertical profiles were identified. The first type exhibited a decrease in wood density with increasing tree height, with the highest wood density observed in the bottom part. This pattern was observed in species such as *Pinus sylvestris*, *Abies alba*, and *Betula pendula*. In contrast, the second type

showed an increase in wood density with tree height, with the highest wood density observed in the top part. This pattern was observed in *Picea abies*, *Larix decidua*, *Quercus robur*, and *Alnus glutinosa* species. Note that a multiple comparison test was conducted for each tree to assess the significance of wood density differences among the bottom, middle, and top discs. Most trees exhibited consistent vertical variation profiles in wood density, except for *Fagus sylvatica*, where both "Top is the highest" and "Bottom is the highest" profiles were observed (Figure 5b).


*Radial variations within the trees*

Unlike the vertical variation, as shown in Figure 6, the radial profiles of wood density for different species exhibit a certain similarity. Across species, wood density tends to decrease from the outer to inner zones, indicating that wood density near the bark is generally higher than that in the center of discs. The magnitude of radial variations in wood density is typically larger

at the bottom disc (represented by brown curves) compared to the middle and top discs (represented by orange and green curves). However, there are exceptions observed in *Quercus robur* and *Fagus sylvatica* species. *Quercus robur* species exhibits an opposite radial profile, with higher wood density in the center than near the bark (Figure 6f). Additionally, *Fagus sylvatica* species is the only species that does not show significant radial variation in wood density (Figure 6e). Furthermore, we conducted a comparison of radial variations between the northern and southern parts of the discs. Using a two-sample t-test,



we found that wood density samples from the northern and southern parts generally do not exhibit significant differences
(Figure 7).

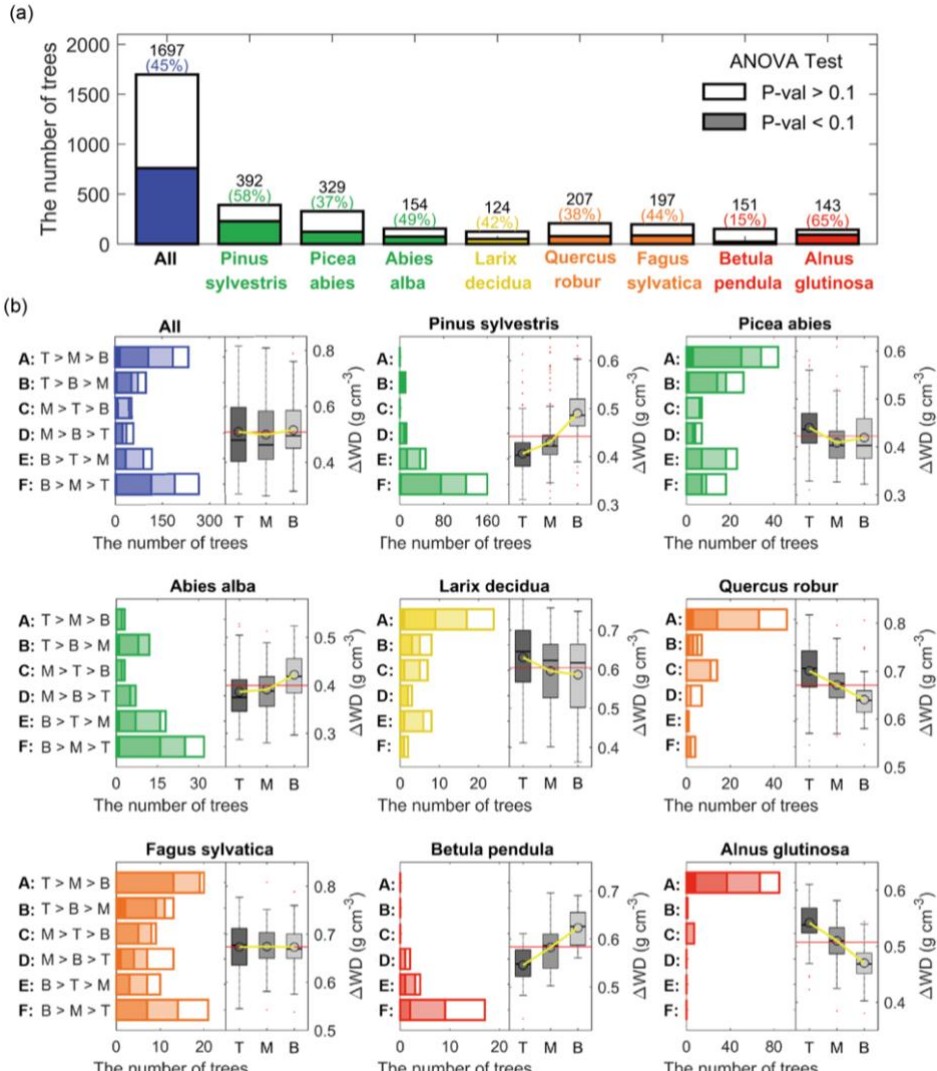

**Figure 5: Comparison of wood density of the discs from the top (T), middle (M) and bottom (B) part of tree. (a) Fractions (The number) of trees with significant differences (P-val < 0.1) in wood density among three levels (top, middle, and bottom) discs via ANOVA analysis. (b) The relation among the top, middle and bottom discs, in comparison to the mean values of wood density, via the multiple comparison test. The color depth in the left-hand panels presents the pairs of two discs with significant difference. The darkest colors indicate that any two of three discs have significant difference (P-val < 0.1), while the lightest colors indicate that none of two discs have significant difference. The right-hand panels show the boxplots of wood density of top, middle, bottom discs of trees, respectively. The yellow curves indicate the changes in the averaged wood density among top, middle and bottom discs. And the red lines present the averages of all wood density records.**






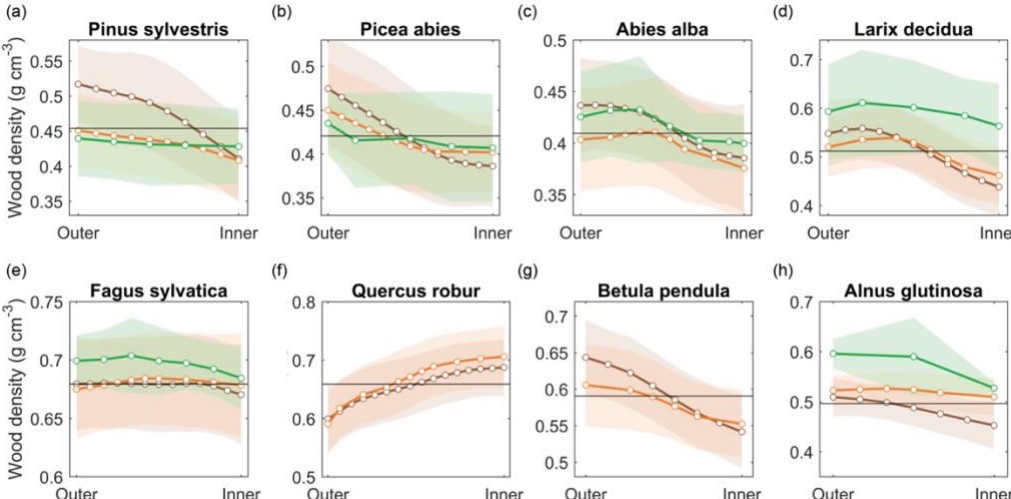

**Figure 6: Radial variations in wood density of top (green), middle (orange) and bottom (brown) discs for eight species. Black line indicates mean wood density. The x-axis indicates relative radial position within the discs (left: pith, right: bark). The colored curves present the mean of wood density, and colored shadings show the one standard deviation of wood density.**

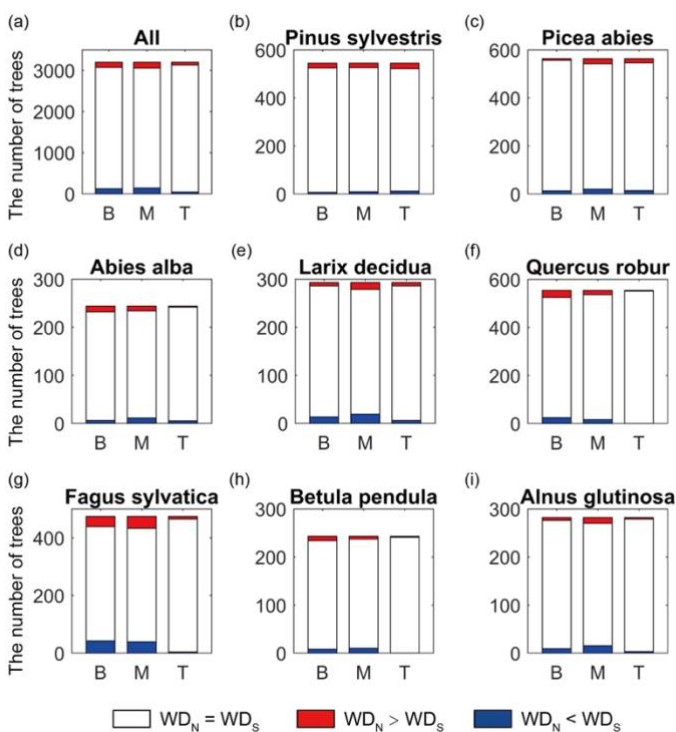

**Figure 7: Comparison in wood density between the northern (WDN) and southern (WDS) part of discs for top, middle and bottom discs respectively. Bar plots show the number of trees, which have non-significant difference between WDN and WDS (white), WDN significantly larger than WDS (red), or WDN significantly smaller than WDS (blue), using t-test at 0.05 significance level. (a) all trees, and (b)-(i) eight tree species separately.**





## 4 Discussions

*Factors influencing the inter-tree variations*

Wood density variation at the large-scale has been reported to relate to climatic variability (Wiemann et al. 2002; Thomas et
al. 2007). However, in this study, the tree-level variations in wood density are more closely related to satellite-based vegetation
indexes rather than climatic variables. This is likely due to the limited heterogeneity of climate variables in the forests of
Poland, where all the wood density measurements were taken. As a result, wood density variations depend more on vegetation
properties, such as tree species, leaf type, and leaf habit. Interestingly, the effects of tree species in explaining spatial variability
in wood density can be substituted by vegetation water and carbon content. Thus, the use of satellite-based NDVI and NDWI
can predict a significant portion of the variations in wood density. In accordance with previous research, our findings align
with the notion that species characterized by tall height require wider vessels to facilitate hydraulic conductivity and sap
transport to leaves, resulting in higher vegetation water content (Coomes et al. 2008), but lower wood density. Conversely, the
relationship between wood density and leaf properties exhibits greater complexity. Earlier studies have reported a negative
association between wood density and leaf size, as well as photosynthetic capacity (Santigo et al. 2004; Wright et al. 2007),
owing to the growth strategies of species with larger leaves, which exhibit faster volumetric growth (Wright et al. 2004).
However, our findings reveal a different relationship between satellite-based vegetation index (NDVI, which usually indicate
canopy greenness and cover) and wood density when controlling for factors such as vegetation water content, landform types,
and soil texture.

Besides climate, we identified the significance of topography in explaining wood density, consistent with findings from other
regional analyses. However, it is important to note that the influence of topography on wood density can vary across different
regions. For example, Sungpalee et al. (2009) found that tree-level wood density in a Thai tropical forest is lower at higher
elevations and on eastern slopes, while Kraft et al. (2008) reported that high tree-level wood density in a Costa Rican montane
forest was found on ridges. These discrepancies can be attributed to the fact that topography may be associated with variations
in soil fertility or light availability. Specifically, valleys tend to have less fertile soils, while trees on ridges may receive more
sunlight compared to those in valleys. The limiting factors for vegetation growth can differ across regions, leading to diverse
relationships between wood density and topography, as well as soil fertility. Our results demonstrate contrasting effects of soil
fertility on wood density across different tree species. Further studies are required to elucidate the underlying causal processes
that contribute to the observed association between wood density and topography, particularly in relationship with factors such
as soil fertility and light availability.

*Factors influencing the intra-tree variations*

Two distinct vertical profiles of wood density may be associated with different tree growth conditions and strategies. The first
profile is characterized by higher wood density in the bottom part of the tree compared to the top part. Trees exhibiting this



profile often inhabit challenging and harsh environmental conditions, such as areas prone to extreme weather events like heavy storms, rains, and snowfall. These trees tend to adopt a conservative growth strategy, prioritizing investments in wood structure (i.e., the bottom part of the tree) over rapid growth (Wright et al., 2004). In contrast, the second vertical profile shows higher wood density in the top part of the tree compared to the bottom part. Trees with this profile tend to adopt a fast-growing, productive strategy, allocating more carbon to the upper regions of the tree to outcompete neighboring trees for essential

resources such as light, water, and nutrients. The mean height of different tree species further supports this hypothesis. For instance, pine trees and fir trees (*Alnus glutinosa* and *Pinus sylvestris* species), exhibiting the first vertical profile, are relatively short, reaching maturity at heights of about 12 to 18 meters, while alder, spruce, oak, and larch trees (*Picea abies*, *Larix decidua*, *Quercus robur*, and *Alnus glutinosa* species) with the second vertical profile are larger, attaining heights of 30 to 50 meters.


The radial profile of wood density commonly exhibits an increase from the center of the tree towards the bark. This pattern is likely attributed to the fact that the carbon uptake capacity of trees tends to increase with tree age and size, resulting in higher wood density in the newly formed growth rings (Thomas and Malczewski 2007). However, *Quercus robur* species deviates from this general radial profile due to its specific cellular structure. The species is characterized by a significant presence of

large vessels in the outer zones, closer to the bark, leading to a lower wood density. This phenomenon is commonly observed in most ring-porous hardwood species (Woodcock and Shier 2002).

*Comparison the magnitude of inter- and intra-tree variation in wood density*

We conducted a comparison between inter- and intra-tree variations in wood density, focusing on the magnitude of these

variations. To quantify the magnitude, we used the normalized standard deviation, also known as the coefficient of variation. The comparison was conducted at both the species level (Figure 8a) and the plot level (Figure 8b). Across all eight species analyzed, the results consistently demonstrate that t the variations within individual trees exhibit larger magnitudes compared to the inter-tree variations between trees. This can be observed in Figure 8, where the data points consistently lie above the 1:1 line. Specifically, the within-tree variations are approximately 1.2 times greater than the inter-tree variations. This finding

emphasizes the significance of understanding the variations in wood density within individual trees. It also suggests that relying on a single sample to represent the wood density of a tree may lead to substantial uncertainty. Also, it is important to note that our findings are based on wood density measurements conducted in Poland. To generalize and validate the comparison between inter- and intra-tree variations in wood density on a larger scale, further investigations are needed.

**5 Conclusion**

Our study, conducted in Poland and focusing on wood density measurements, investigates the variations in wood density both between and within trees. Our results suggest that significant differences in wood density measurements among different tree species. Through the implementation of a random forest model, we demonstrated that the combined use of satellite-based





vegetation indexes (such as NDVI and NDWI), topographic variables, and soil sand content can effectively predict 91% of the
inter-tree variations in wood density. Furthermore, within individual trees, we observed variations in wood density across
different tree heights and along the radial direction from the inner to the outer zones of the discs. These vertical and radial
profiles of wood density within trees may be attributed to climatic conditions, growth strategies, and the physiological structure
of the trees. Notably, we found that the magnitude of wood density measurements within trees is substantial, even surpassing
the magnitude of inter-tree variations in wood density. This emphasizes the significance of considering the intra-tree variations
when analyzing wood density.

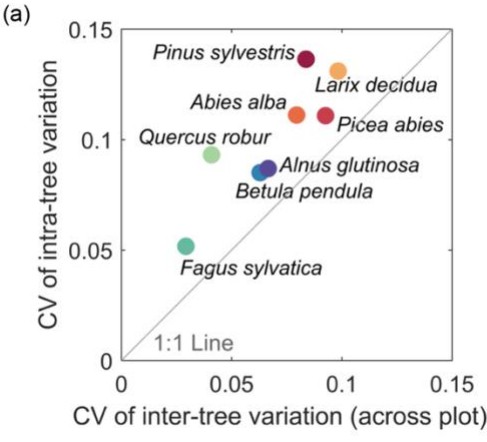
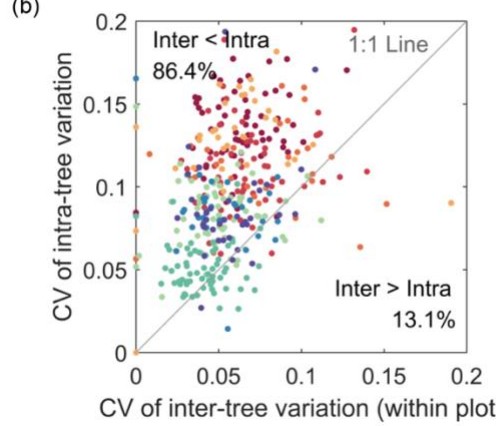

**Figure 8. (a) Comparison of the magnitude of inter-tree variations in wood density across plots (within a species) and intra-tree variations. Both inter- and intra-tree variations were quantified using standard deviation of wood density, and were normalized by the mean values. (b) Comparison of the magnitude of inter-tree variations in wood density within a plot and intra-tree variations. The color of dots presents the species of trees within the plot, same as panel (a).**


**Acknowledgements**

The data used for analysis were collected under REMBIOFOR project entitled "Remote sensing-based assessment of woody
biomass and carbon storage in forests", which was financially supported by the National Centre for Research and Development
(Poland), under the BIOSTRATEG programme (Agreement No. BIOSTRATEG1/267755/4/NCBR/2015). H. Y. is supported
by the Project Office BIOMASS (grant number 50EE1904) funded by the German Federal Ministry for Economic Affairs and
Climate Action.

**Data Availability Statement**

All published data sources have been referenced in the manuscript. All the data supporting the findings of this study will be
publicly available after acceptation.



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
