# Peer review of "Similar importance of inter-tree and intra-tree variations in wood density observations in Central Europe"

_EGUsphere, 2023_

## Referee Comment (RC1)

**Review for Yang et al. 2023's paper**

'Similar importance of inter-tree and intra-tree variations in wood density observations in Central Europe'

Using a feature selection approach and random forest model, the study conducted by Yang et al. identify key predictors of wood density, including satellite-based vegetation indexes, topographic variables, and soil sand content. The database used is both important and impressive. It offers a robust statistical perspective on wood density. However, the paper has some weaknesses that require partial rewriting. The methodology is insufficiently presented, with some factors (e.g., NDVI, topographic metrics) appearing in the results section without proper introduction or details about their origin and precision. Some elaborate statistics are not and should be introduced in the methods section. In the results section, the statistical significance of the results should be systematically presented. The discussion section is short and miss some important aspects, like the relevance and bias of the sampling which are barely addressed. The authors do not place their findings in a broader perspective and, in particular, do not clearly highlight what aligns with what is already known versus what is new. They may discuss the implications of their results on the understanding and modeling of the carbon cycle. Some parts of the paper (introduction and discussion, in particular) are not very well written. A careful reading and corrections are necessary. Some references should be added. Some are cited in the text but missing in the references list.

In conclusion, the paper deserves to be published, but only after addressing these issues and being partly rewritten.

**Detailed comments:**

Throughout the paper, the figures should appear after being called.

**Introduction:**

Line 30: ...is a fundamental trait which describes the carbon...
Line 31: a key indicator for various ecological and physiological processes, such as: you cannot say that hydraulic properties is an example of an ecological or physiological process.
Line 36: has the potential to have a positive impact: consider : may impact positively the ...
Line 38: ...Kraft et ak. 2010). For example, Chao...
Line 39: Chao et al. 's paper shows that tree mortality rates are related to wood density but, to my understanding, not due to wood density.
Line 50: 'It is important to note that these influencing factors can vary from one region to another'. It is so obvious that I do not think it is worth mentioning...
Line 52: ...and lower elevation (Sungpalee et al. 2009) and the soil water availability, which has a positive impact on wood density in dry biomes and a negative impact in wet biomes (Rocha et al. 2020).
Line 55: I would rather say : 'fast growth due to low competition for light and space' . Gryc (and not Gyrc) et al 2011 is missing from the list of references. I guess it is : DOI: 10.5194/egusphere-2023-2691
You should make clear that this conclusion was reached for coniferous trees.
Line 57: How does the elements above suggest anything about tree development stages?
Line 60: 'a novel dataset of wood density measurements'
Line 66: northern or southern discs? How can a disk have an orientation?

Line 68: 'Compare the extent of inter-tree and intra-tree variation in wood density for the tree species or forest plots analysed'.
Line 71: What do you mean by 'wood density formulas'?
'only the older trees'
Line 72: 'In many previous works': references are needed.
Line 73: What do you mean: 'The relationships between 'the changes in environmental conditions and variations in wood density' or 'the relationship between environmental conditions and wood density'?
Line 74: 'collected in Poland'

**Methods**

Line 79: 'wood density samples...': I suggest: 'the density measurements of more than 48000 samples, from 2920 trees from 391 forest plots in Poland, carried out....'
Line 85: fertile/ low fertile soil: On what criteria is this separation based?
Line 87: What is the typical elevation of low and high plots?
Line87:'...add more complexity': I would not say it is more complex. These are just additional criteria worth exploring.
Line 90: 'analysis belong to three Plant functional types (PFTs) ..'
Line 94: 'Regarding the age of the trees, all trees can be classified into nine age classes: I suggest: 'The tree population were divided arbitrarily into nine age clases'
Line 95: 'bottom of a trunk': What height?
Line 96 and fwd: please replace 'can be classified' by 'were classified'.
Line 97 and 98 : The height and DBH of trees are not 'defined as '. you may consider 'divided into the following categories'

Figure 1: What about: 'a) Distribution of samples (solid colour bars), trees (light colour bars), and forest plots (transparent bars) utilized for density measurements across the eight species; b) location of the 391 forest plots in Poland (symbols as in panel a).

Line 103: The sampling procedure is unclear. You mention taking samples with an increment borer and then refer to discs. Do you mean that you took cores from the discs? If so, please make it clear.

Line 106: To my understanding: 'bottom", 'middle' and "top" can correspond to very different height depending on the total height of the tree. Therefore are these categories meaningful?

Line 107: What do you mean here by **sampling** (the sampling was conducted from the inner ...'. Do you mean the density measurements? How was the wood density determined? The proocedure used should be described in the method section. The data should be made available in a repository, along with supplementary materials that include relevant statistics (e.g., mean diameter, height, elevation for each species) and detailed information about vegetation indexes, water content, etc.

Line 115-fwd: The trees have varying diameters, and some species are more represented than others (e.g., Pinus sylvestris versus Alnus glutinosa). Therefore, is the average wood density truly representative of the population? If not, does it matter? Could this be corrected by using a weighted mean? These points should be presented and discussed.

Line 123: 'Less than 100 or 500 m, grid sizes of 0.05 or 0.1°'. The results indicate no consistent differences when comparing distances or grid sizes. Therefore, you may explain that both criteria (100 and 500; 0.05 and 0.1) were tested and yielded similar results. For simplicity, you may present the results for only one distance and one grid size.

Line 133: Where is Table 1?? Where do the covariates come from? How reliable are they?

**Results**

Line 141-142: 'For the analysis... 2920 trees": Already said in the method section.
Lie 145: '... is lower  than the density of...'
Line 148: '...slightly lower' : Is it statistically significant?
...'significantly lower': You should systematically test the significance of the differences between means.
Figure 2: 'Eight tree species... "unnecessary as it is stated in the text.
Line 163: 'the large difference in wood density observed among species tend to diminish when considering the averages within geographical locations'. Or does it show that location is just not a discriminating factor?

Line 165: 'attributed to the relatively even distribution of the eight primary species': I do not interpret it this way, as the species are evenly distributed. If the spatial distribution were uneven, there could be a potential confusion between species and location, but that is not the case here. There is likely no spatial bias, and the small percentage of the variance explained indicates that location is not a predictor for wood density.
Aren't the fertility and elevation criteria more or less included into the location criteria (grid cell)? What is the scale for the difference ? What does the black line represent?

Line 174-fwd: How are height, DBH and age related to one another? They are probably not independent. How do you separate the effect on wood density of height or DBH from age?

Line 177: Taking into account the uncertainties, can you say that there is a difference of behaviour between broadleaf and needleleaf base on the age. In addition, the lack of VIII and IX class for needleleaf trees, makes the statement irrelevant.

Line 179: 'for broadleaf the impact of height is more pronounced for taller trees': what does it mean? The graph indicates that taller trees tend to have a higher average wood density, but the standard deviation is substantial. The significance of the difference between the means, such as between class III and VII, should be evaluated. What are the relevant statistics?

Line 190's: The description of the random forest approach, of the factors tested (including where they come from, and their accuracy) should be provided in the method section.

Line 200: link between NDVI and NDWI values and carbon and water contents respectively: references are needed.

Figure 4: What are the y-axis in b. Is the shap values for density in %?

Line 210: With a $p < 0.1$ the evidence is weak.
Line 221: The multiple comparison test should be presented in the method section

Line 230 'The magnitude of radial variations in wood density is typically larger at the bottom disc compared to the middle and top discs ': The older part of the tree is located in the bottom disk. How might this affect the results?

 'However, ….' Should come before this sentence, as it refers to the outer-inner gradients in the various species.

**Discussion**
Line 259: I propose: 'Large-scale variations in wood density have been reported to correlate with climatic variability. However, in this study, tree-level variations in wood density are more strongly linked to vegetation indices than to climatic variables'.

Line 261: variability rather than heterogeneity?

Line 265: I propose: 'Therefore, the utilization of satellite-based NDVI and NDWI can effectively predict a substantial portion of the variations in wood density.'

Line 268: 'exhibit greater complexity': Than what?

Line 269-fwd: Your point is not very clear. What implicit relationship do you draw between NDVI and leaf size?

Line 289: 'Specifically, valleys tend to have less fertile soils,  In addition, trees on ridges may receive more sunlight compared to those in valleys.'

Line 295: For instance, pine trees and fir trees (*Alnus glutinosa* and *Pinus sylvestris* species), Please give latin name consistent with vernacular ones!

Line 297: 'while alder, spruce, oak, and larch trees (*Picea abies*, *Larix decidua*, *Quercus robur*, and *Alnus glutinosa* species) ': same order for vernacular and latin name, please.

Line 304: 'The species is characterized by a significant presence of  large vessels in the outer zones, closer to the bark, leading to a lower wood density'. Where is this from? Not in Woodcock and Shier, 2002.'These authors rather link radial increase to early successional status (and change of carbon allocation when trees reach the canopy and are subjected to more wind).

Line 311:  Please correct: 'Across all eight species analyzed, the results consistently demonstrate that  the variations within individual trees exhibit larger magnitudes compared to the inter-tree variations '

---

## Author Comment (AC1)

**To Reviewer #1**:

Thank you for your time and effort to review our manuscript. Each request or comment is repeated below in black, and our responses are in blue. Additional or altered text that has appeared in the manuscript is marked in red.

**[General Comment]** Using a feature selection approach and random forest model, the study conducted by Yang et al. identify key predictors of wood density, including satellite-based vegetation indexes, topographic variables, and soil sand content. The database used is both important and impressive. It offers a robust statistical perspective on wood density. However, the paper has some weaknesses that require partial rewriting. The methodology is insufficiently presented, with some factors (e.g., NDVI, topographic metrics) appearing in the results section without proper introduction or details about their origin and precision. Some elaborate statistics are not and should be introduced in the methods section. In the results section, the statistical significance of the results should be systematically presented. The discussion section is short and miss some important aspects, like the relevance and bias of the sampling which are barely addressed. The authors do not place their findings in a broader perspective and, in particular, do not clearly highlight what aligns with what is already known versus what is new. They may discuss the implications of their results on the understanding and modeling of the carbon cycle. Some parts of the paper (introduction and discussion, in particular) are not very well written. A careful reading and corrections are necessary. Some references should be added. Some are cited in the text but missing in the references list. In conclusion, the paper deserves to be published, but only after addressing these issues and being partly rewritten.

**[Response]** Thank you for your constructive and positive comments. We appreciate your efforts to improve our manuscript. We provided answers and clarified all the points that you raised, and revised our manuscript accordingly. Detailed responses can be found below.

**[Detailed comments]** Throughout the paper, the figures should appear after being called

**[Response]** Corrected.

**[Introduction]**:

Line 30: ...is a fundamental trait which describes the carbon...

Line 38: ...Kraft et ak. 2010). For example, Chao...

Line 52: ...and lower elevation (Sungpalee et al. 2009) and the soil water availability, which has a positive impact on wood density in dry biomes and a negative impact in wet biomes (Rocha et al. 2020).

Line 60: 'a novel dataset of wood density measurements'

Line 68: 'Compare the extent of inter-tree and intra-tree variation in wood density for the tree species or forest plots analysed'.

Line 74: 'collected in Poland'

**[Response]** Corrected.

Line 31: a key indicator for various ecological and physiological processes, such as: you cannot say that hydraulic properties is an example of an ecological or physiological process.

**[Response]** The sentence is now written as: "It serves as a key indicator for various ecological and physiological processes, such as tree growth, mortality rates, and the vulnerability to hydraulic failure." (Page: 2; Lines: 40-41)

Line 36: has the potential to have a positive impact: consider : may impact positively the ...

**[Response]** Thanks. The sentence has been revised as you suggested.

Line 39: Chao et al. 's paper shows that tree mortality rates are related to wood density but, to my understanding, not due to wood density.

**[Response]** Thank you for pointing it out. The sentence has been revised as follow: "Chao et al. (2008) found contrasting tree mortality rates in eastern and western Amazon forests, which were related to variations in wood density, with lower density associated with higher mortality rates." (Page: 2; Lines: 47-49)

Line 50: 'It is important to note that these influencing factors can vary from one region to another'. It is so obvious that I do not think it is worth mentioning...

**[Response]** The sentence has been deleted.

Line 55: I would rather say : 'fast growth due to low competition for light and space' . Gryc (and not Gyrc) et al 2011 is missing from the list of references. I guess it is : DOI: 10.5194/egusphere-2023-2691 You should make clear that this conclusion was reached for coniferous trees.

**[Response]** Thank you for your comment and suggestion. The sentence has been revised to: "For example, Gryc et al. (2011) found that in coniferous trees, thin cell walls resulting from fast growth due to lower competition for light and space are typically associated with low wood density. Conversely, thick cell walls, which result from slower growth, are related to high wood density." (Page: 2; Lines: 62-64). Additionally, the reference of Gryc et al. (2011) (Thanks for your correction) has been added to the reference list.

Line 57: How does the elements above suggest anything about tree development stages?

**[Response]** We have re-written this part as follows: "Additionally, the growth rate of individual trees can vary over their lifespan, leading to variations in wood density. Generally, young trees grow quickly while mature trees grow steadily (Bowman et al., 2012). Thus, the growth strategy between trees and the development stage of an individual tree's lifespan can play a role in shaping wood density gradients." (Page: 2; Lines: 65-68)

Line 66: northern or southern discs? How can a disk have an orientation?

**[Response]** We apologize for confusion caused by our previous wording. Each disc was divided into two parts, i.e., the northern and southern part. We have clarified this in the revised manuscript.

Line 71: What do you mean by 'wood density formulas'? 'only the older trees'
Line 72: 'In many previous works': references are needed.

**[Response]** Sorry for our overly ambitious sentences. We have rewritten this part as follows: "Many previous studies have assessed the relationship between environmental conditions and wood density using data from limited the mature forest plots (Baker et al 2004; Dias et al 2018; Phillips et al 2019)." (Page: 3; Lines: 126-127)

Line 73: What do you mean: 'The relationships between 'the changes in environmental conditions and variations in wood density' or 'the relationship between environmental conditions and wood density'?

**[Response]** We have revised the sentence as "the relationship between environmental conditions and wood density." (Page: 3; Line: 126)

**[Methods]**

Line 79: 'wood density samples...': I suggest: 'the density measurements of more than 48000 samples, from 2920 trees from 391 forest plots in Poland, carried out....'

Line 90: 'analysis belong to three Plant functional types (PFTs) categories..'

Line 94: 'Regarding the age of the trees, all trees can be classified into nine age classes: I suggest: 'The tree population were divided arbitrarily into nine age clases'

Line 96 and fwd: please replace 'can be classified' by 'were classified'.

Line 97 and 98 : The height and DBH of trees are not 'defined as '. you may consider 'divided into the following categories'

[Response] Thanks. These sentences have been revised as you suggested.

Line 85: fertile/ low fertile soil: On what criteria is this separation based?

Line 87: What is the typical elevation of low and high plots?

[Response] Thank you for your comment. We have clarified this in the revised manuscript: "Low plots are typically located in lowlands, ranging from 0 to 300 m asl. In this study, field plots were selected from elevations not exceeding 100 m asl. In contrast, high plots begin at 300 m asl, encompassing both uplands and mountainous areas. In Polish conditions, these high plots extend up to 1600 m asl." (Page: 3; Lines: 142-145)

Line87:'...add more complexity': I would not say it is more complex. These are just additional criteria worth exploring.

[Response] The sentence has been revised as follows: "These distinctions within the species-specific plots allow for a more detailed exploration of the impacts of environmental conditions on wood density." (Page: 4; Lines: 193-194)

Line 95: 'bottom of a trunk': What height?

[Response] Thank you for your question. Unfortunately, we cannot provide the exact height from the base of the trunk where the rings were counted to determine the tree's age. According to the project protocol, each tree was divided into three equal parts. Discs were cut from the middle section of these three logs for wood density measurements. This approach ensures consistency in the biological and physico-mechanical properties of wood samples, regardless of the tree's height or age. The final disc for counting annual rings was cut from the bottom section of the lower log. The height at which this disc was taken depended on the tree's diameter and safety considerations during felling with a chainsaw.

Figure 1: What about: 'a) Distribution of samples (solid colour bars), trees (light colour bars), and forest plots (transparent bars) utilized for density measurements across the eight species; b) location of the 391 forest plots in Poland (symbols as in panel a).

**[Response]** We has revised the caption of Figure 1 as you suggested.

Line 103: The sampling procedure is unclear. You mention taking samples with an increment borer and then refer to discs. Do you mean that you took cores from the discs? If so, please make it clear.

**[Response]** Yes, thanks for pointing it out. We now make it clear in the revised manuscript.

Line 106: To my understanding: 'bottom", 'middle' and "top" can correspond to very different height depending on the total height of the tree. Therefore are these categories meaningful?

**[Response]** The reviewer is correct that the 'bottom', 'middle' and 'top' categories correspond to different heights for different trees, making direct comparison across trees less meaningful. However, our approach was to compare the wood density between 'bottom', 'middle' and 'top' categories for each individual tree (as shown in Figure 5). We believe these intra-tree comparisons are valid, and using the terms 'bottom', 'middle' and 'top' provides readers with an intuitive understanding of the vertical profile of wood density within each tree.

Line 107: What do you mean here by sampling (the sampling was conducted from the inner ...'. Do you mean the density measurements? How was the wood density determined? The proocedure used should be described in the method section. The data should be made available in a repository, along with supplementary materials that include relevant statistics (e.g., mean diameter, height, elevation for each species) and detailed information about vegetation indexes, water content, etc.

**[Response]** Thank you for your comments.

Firstly, we have added the detail of procedures for collecting wood density measurements in the revised manuscript, as follows: "Each disc was cut from north to south to obtain a strip of wood. The samples were divided and numbered into two rays: north and south, starting from the core to the peripheral part. This method allowed for the estimation of variation in the radial density of wood. The number of samples obtained for each disc varied depending on the width of the disc, but each disc typically yielded more than 10 samples along these radial directions. Standardized wood density samples, measuring 2×2×3 cm, were cut from the strips, which were dried in a dryer at temperature of $103 \pm 2$ ℃ to an absolutely dry state. After the samples

cooled down in the desiccator, the linear dimensions of the samples were measured using an electronic caliper, and their weight was measured on a laboratory scale. The stereometric density was then calculated from the classical mass/volume formula." (Page: 5; Lines: 232-241)

Secondly, regarding making the data publicly available, we respectfully request a delay in its release. Our colleagues in Poland have invested significant effort in building this complex database and would like to retain priority for its use. There are ongoing research projects based on this database, and once those are published, the data will be made available in a public repository.

Thirdly, while we cannot make the full wood density database public, we have added a new table in the supplementary materials that includes the mean, minimum, maximum, 25th, and 75th percentiles of wood density for each species.

**Table S1**. The mean, minimum, maximum, and the 25th and 75th percentiles of wood density for eight tree species.

| WD (g cm$^{-3}$) | Species | | | | | | | |
|---|---|---|---|---|---|---|---|---|
| | *Pinus sylvestris* | *Picea abies* | *Abies alba* | *Larix decidua* | *Quercus robur* | *Fagus sylvatica* | *Betula pendula* | *Alnus glutinosa* |
| Min | 0.31 | 0.29 | 0.30 | 0.32 | 0.45 | 0.57 | 0.45 | 0.38 |
| $Q_{25}$ | 0.41 | 0.38 | 0.38 | 0.46 | 0.64 | 0.66 | 0.55 | 0.47 |
| Mean | 0.44 | 0.41 | 0.40 | 0.50 | 0.67 | 0.68 | 0.57 | 0.49 |
| $Q_{75}$ | 0.47 | 0.44 | 0.43 | 0.54 | 0.70 | 0.70 | 0.60 | 0.52 |
| Max | 0.58 | 0.56 | 0.63 | 0.62 | 0.80 | 0.78 | 0.68 | 0.62 |

Line 115-fwd: The trees have varying diameters, and some species are more represented than others (e.g., Pinus sylvestris versus Alnus glutinosa). Therefore, is the average wood density truly representative of the population? If not, does it matter? Could this be corrected by using a weighted mean? These points should be presented and discussed.

**[Response]** Good point. Thank you for this very constructive suggestion.

As the reviewer mentioned, we calculated the average of sample-level wood density to represent the tree-level wood density. Firstly, to determine if the average value is a good representation of population, we compared the mean and median values within each individual trees. As shown in Figure R2a, the difference between mean and median values are minor (slope: 1.02, $R^2 = 0.99$). This suggests that the wood density samples within individual tree tend to follow a normal distribution, and the tree-level wood density can be effectively represented by the average of the sample-level values. Secondly, we examined whether the intra-tree variability of wood density differs among tree species. Figure R2b shows that, despite varying tree diameters, the magnitude of intra-tree variability is similar across tree species, with no significant differences observed.

[Figure]

**Figure R1**. (a) Comparison of the mean and median values of wood density within the individual trees. (b) The standard deviation (STD) of the sample-level wood density within the individual trees. Boxplots indicate the median, mean, minimum, maximum, and the 25th and 75th percentiles of wood density for eight tree species.

We also added the following text: "Tree-level wood density was calculated by the average of all samples within each individual trees. This method is used because there was no significant difference between the mean and median values of the samples (Figure S1a), indicating that wood density within an individual tree typically follows a normal distribution. Furthermore, the magnitude of intra-tree variability is consistent across eight tree species (Figure S1b)." (Page: 6; Lines: 274-277)

Line 123: 'Less than 100 or 500 m, grid sizes of 0.05 or 0.1°'. The results indicate no consistent differences when comparing distances or grid sizes. Therefore, you may explain that both

criteria (100 and 500; 0.05 and 0.1) were tested and yielded similar results. For simplicity, you may present the results for only one distance and one grid size.

**[Response]** We has revised the related texts in the Method and Results as you suggested.

Line 133: Where is Table 1?? Where do the covariates come from? How reliable are they?

**[Response]** We apologize for the oversight. Table 1 has been added, and the sources of covariates are now listed. We have verified that none of these data have reported data quality issues in Central Europe.

**Table 1**. The predictor covariates used in the random forest model for inter-tree variations in wood density. The original 8-daily values of NDVI and NDWI were aggregated into a median (P50) and a standard deviation (STD) for the entire period.

| Variables | Description | Unit | Original resolution | Source |
|---|---|---|---|---|
| SNDPPT | Weight percentage of the sand particles (0.05–2 mm) | % | 250 m | SoilGrids database |
| NDWI | 8-daily Enhanced Vegetation Index (EVI) generated using the gridded daily surface reflectance product. | 1 | 0.083º | MOD13A2 |
| NDVI | 8-daily Normalized Difference Vegetation Index (NDVI) generated using the gridded daily surface reflectance product. | 1 | | |
| Geomorphons | a pattern recognition approach to classification and mapping of landforms from digital elevation models (DEMs) | - | 30m | Jasiewicz & Stepinski (2013) |

**[Results]**

Line 141-142: 'For the analysis... 2920 trees": Already said in the method section.

**[Response]** The sentence has been deleted.

Line 145: '... is lower compared to than the density of...'

Figure 2: 'Eight tree species... "unnecessary as it is stated in the text.

**[Response]** Corrected.

Line 148: '...slightly lower' : Is it statistically significant? ...'significantly lower': You should systematically test the significance of the differences between means.
**[Response]** We have added the significance of differences in between tree families.

Line 163: 'the large difference in wood density observed among species tend to diminish when considering the averages within geographical locations'. Or does it show that location is just not a discriminating factor?
Line 165: 'attributed to the relatively even distribution of the eight primary species': I do not interpret it this way, as the species are evenly distributed. If the spatial distribution were uneven, there could be a potential confusion between species and location, but that is not the case here. There is likely no spatial bias, and the small percentage of the variance explained indicates that location is not a predictor for wood density. Aren't the fertility and elevation criteria more or less included into the location criteria (grid cell)? What is the scale for the difference ? What does the black line represent?
**[Response]** Firstly, we agree with the reviewer that location is not a discriminating factor, and we have clarified this in the revised manuscript. "The reason is that the location does not differentiate the tree species in Poland; as different tree species with varying wood density are distributed across similar locations in Poland, as shown in Figure 1b." (Page: 8; Lines: 332-333)

Secondly, we agree with the reviewer's insightful point. It is indeed correct that soil, elevation, and vegetation properties can account for the variation not explained by location. This is why we found that the random forest model, even without species information, still achieved a high $R^2$ value (0.91) when using only these vegetation, soil, and topography-related covariates. We have added the sentence in the revised manuscript: "This variance in wood density, unexplained by location, could be related to local environmental conditions such as vegetation characteristics, soil properties and topography." (Page: 8; Lines: 333-335)

Additionally, the black line is used to bracket the labels related to location.

Line 174-fwd: How are height, DBH and age related to one another? They are probably not independent. How do you separate the effect on wood density of height or DBH from age?

**[Response]** We agree with the reviewer that height, DBH and age are inter-related and not independent variables. Generally, height and DBH increase with age, especially in the younger trees, while the growing rates of mature trees are relatively low. Since the combined contribution of tree age, height, and DBH to wood density variance are very low (less than 6%), we don't believe it is necessary to separate their individual contribution. Nevertheless, in the revised manuscript, we clarify that the collinearity among height, DBH and age may result in an overestimation of the contribution of each individual factor, as follow: "Note that height, DBH and age are interrelated and not independent variables, and their collinearity may result in an overestimation of the contribution of each individual factor." (Page: 9; Lines: 360-3631)

Line 177: Taking into account the uncertainties, can you say that there is a difference of behaviour between broadleaf and needleleaf base on the age. In addition, the lack of VIII and IX class for needleleaf trees, makes the statement irrelevant.

**[Response]** Thanks for pointing it out. We have rewritten the related texts as follows: "regarding the impacts of tree ages (Figure 3a), for broadleaf trees, wood density tends to increase with tree age up to approximately 140 years (class VII), after which it stabilizes. For needleleaf trees, wood density also exhibits an increase in wood density with age up to 140 years (class VII). Since there are no observation of wood density from older needleleaf trees, it remains unclear whether wood density would continue increase or stabilized beyond 140 years." (Page: 9; Lines: 362-366)

Line 179: 'for broadleaf the impact of height is more pronounced for taller trees': what does it mean? The graph indicates that taller trees tend to have a higher average wood density, but the standard deviation is substantial. The significance of the difference between the means, such as between class III and VII, should be evaluated. What are the relevant statistics?

**[Response]** Our previous statement was mainly based on the mean values for each classes, and we understand the reviewer concerns on the no difference in standard deviation. We have rewritten the related texts as follows: "both needleleaf and broadleaf trees show an increase in mean wood density with height and DBH classes, especially for tall broadleaf trees (height class $\geq 4$, i.e., tree height $\geq 20$m). However, the variance in wood density within these height and DBH classes are large, resulting in no statistically significant difference in wood density distributions among classes (Figure 3b-c)." (Page: 9; Lines: 366-370)

Line 190's: The description of the random forest approach, of the factors tested (including where they come from, and their accuracy) should be provided in the method section.

**[Response]** These points related to the random forest model have been added into the method section.

Line 200: link between NDVI and NDWI values and carbon and water contents respectively: references are needed.

**[Response]** The references have been added.

Figure 4: What are the y-axis in b. Is the shap values for density in %?

**[Response]** The y-axis in Figure 4b represents the absolute values of wood density, measured in g cm$^{-3}$. This has been clarified in the caption of Figure 4.

Line 210: With a p<0.1 the evidence is weak.

**[Response]** We re-do the analysis using a threshold of $p < 0.05$ for significance, and the updated results are generally consistent with the previous findings. We have also revised the corresponding text accordingly.

The corresponding texts have been revised as follow: "To examine the differences in wood density among the bottom, middle and top parts of trees, we conducted an ANOVA analysis. The results indicate that about 35% of all the trees exhibit significant vertical variations (p-value $< 0.05$) in wood density (Figure 5a). The percentage of trees with significant vertical variations (p-value $< 0.05$) varied among species, with Alnus glutinosa and Pinus sylvestris species having the highest percentage (around 50%), while Betula pendula species had the lowest percentage of 8%." (Page: 11; Lines: 408-412)

[Figure]

**Figure 5**. Comparison of wood density of the discs from the top (T), middle (M) and bottom (B) part of tree. (a) Fractions (The number) of trees with significant differences (P-val < 0.05) in wood density among three levels (top, middle, and bottom) discs via ANOVA analysis. (b) The relation among the top, middle and bottom discs, in comparison to the mean values of wood density, via the multiple comparison test. The color depth in the left-hand panels presents the pairs of two discs with significant difference. The darkest colors indicate that any two of three discs have significant difference (P-val < 0.05), while the lightest colors indicate that none of two discs have significant difference. The right-hand panels show the boxplots of wood density of top, middle, bottom discs of trees, respectively. The yellow curves indicate the changes in the averaged wood density among top, middle and bottom discs. And the red lines present the averages of all wood density records.

Line 221: The multiple comparison test should be presented in the method section

**[Response]** Thanks for pointing it out. We have moved the text related the multiple comparison test into the method section.

Line 230 'The magnitude of radial variations in wood density is typically larger at the bottom disc compared to the middle and top discs ': The older part of the tree is located in the bottom disk. How might this affect the results?

'However, ….' Should come before this sentence, as it refers to the outer-inner gradients in the various species.

**[Response]** Thank you so much for your comments. We think this pointing raised by the reviewer is a possible explanation for the large magnitude of radial variations at the bottom discs. We have added the sentence: "This greater variation at the bottom discs could be associated with the presence of older and earliest growing parts, which are only located in the bottom sections of the trees." (Page: 11; Lines: 429-430). Additionally, the word of 'However' has been deleted.

**[Discussion]**

Line 259: I propose: 'Large-scale variations in wood density have been reported to correlate with climatic variability. However, in this study, tree-level variations in wood density are more strongly linked to vegetation indices than to climatic variables'.

**[Response]** Thanks. These sentences have been revised as you suggested.

Line 261: variability rather than heterogeneity?

Line 268: 'exhibit greater complexity': Than what?

**[Response]** Corrected.

Line 265: I propose: 'Therefore, the utilization of satellite-based NDVI and NDWI can effectively predict a substantial portion of the variations in wood density.'

**[Response]** Thanks. These sentences have been revised as you suggested.

Line 269-fwd: Your point is not very clear. What implicit relationship do you draw between NDVI and leaf size?

**[Response]** We have rewritten this part as follows: "However, our findings reveal an opposite relationship between satellite-based vegetation index (NDVI, which usually indicate canopy greenness and cover) and wood density when controlling for factors such as vegetation water content, landform types, and soil texture. Specifically, trees with high NDVI (indicating large canopy coverage) exhibit high wood density." (Page: 14; Lines: 488-490)

Line 289: 'Specifically, valleys tend to have less fertile soils. In addition, trees on ridges may receive more sunlight compared to those in valleys.'

**[Response]** Corrected.

Line 295: For instance, pine trees and fir trees (Alnus glutinosa and Pinus sylvestris species), Please give latin name consistent with vernacular ones!

Line 297: 'while alder, spruce, oak, and larch trees (Picea abies, Larix decidua, Quercus robur, and Alnus glutinosa species) ': same order for vernacular and latin name, please.

**[Response]** Thanks for pointing it out. We have corrected all of vernacular names.

Line 304: 'The species is characterized by a significant presence of large vessels in the outer zones, closer to the bark, leading to a lower wood density'. Where is this from? Not in Woodcock and Shier, 2002.'These authors rather link radial increase to early successional status (and change of carbon allocation when trees reach the canopy and are subjected to more wind).

**[Response]** We have added the reference.

Line 311: Please correct: 'Across all eight species analyzed, the results consistently demonstrate that the variations within individual trees exhibit larger magnitudes compared to the inter-tree variations between trees.'

**[Response]** Corrected.

---

## Author Comment (AC2)

**To Reviewer #2**:

Thank you for your time and effort to review our manuscript. Each request or comment is repeated below in black, and our responses are in blue. Additional or altered text that has appeared in the manuscript is marked in red.

**[General Comment]** This paper utilizes an impressive dataset of 48,000 wood density samples from forest in Poland to investigate within and among tree variations in wood density. Taxonomic and landscape factors are correlated with tree density, then a feature selection and random forest approach is used to model spatial variation in wood density using remote sensing metrics.

While I think this dataset and findings are interesting, there is some disconnect between the objectives and analyses, and the analytical approach (and interpretation therein) needs improvement. Moreover, the introduction and discussion are both lacking in depth of narrative development and interpretation and lacks a thorough review of the wide body of literature that has focused on wood density variability.

**[Response]** Corrected.

**[Detailed comments]** For this review, however, I will focus primarily on the analytical approach, as I think corrections must be made here first before the interpretation of results in the discussion can be evaluated.

1. Were density samples collected to minimize inclusion of compression or tension wood? Xylem cells can develop strong differences in cell wall thickness (and thus density) due to directional effects of topography/wind speed which could bias density measurements and possibly inflate within-tree variability. It's mentioned that rings were sampled for radial profiles in north and south directions, so it seems likely that some compression or tension wood has been included and possibly influencing results. This needs clarification.

**[Response]** Thank you for this comment. During the measurement process, each ensity sample was examined for wood defects, including compression and tension wood, knots, resin wood, cracks, abnormal shapes after drying, and other defects. For the presented research samples without any defects have been taken for further analysis due to the standard of small samples density measurement. However, there is still a very interesting base of complete samples with defects influencing the "real" wood density which one can measure with the whole discs without removing and cutting off any wood defects. We have clarified this in the revised

manuscript, as follows: "Each disc was cut from north to south to obtain a strip of wood. The samples were divided and numbered into two rays: north and south, starting from the core to the peripheral part. This method allowed for the estimation of variation in the radial density of wood. The number of samples obtained for each disc varied depending on the width of the disc, but each disc typically yielded more than 10 samples along these radial directions. Standardized wood density samples, measuring 2×2×3 cm, were cut from the strips, which were dried in a dryer at temperature of $103 \pm 2$ °C to an absolutely dry state. After the samples cooled down in the desiccator, the linear dimensions of the samples were measured using an electronic caliper, and their weight was measured on a laboratory scale. The stereometric density was then calculated from the classical mass/volume formula. During measurement, each density sample was examined for wood defects such as compression and tension wood, knots, resin wood, cracks, abnormal shapes after drying, and other irregularities. In this study, only defect-free samples were selected for further analysis, adhering to the standards for small sample density measurement." (Page: 5; Lines: 232-241)

2. A better description of the predictor data is needed. Line 133 references Table 1, but there are no tables present in the version of this manuscript that I reviewed or in the supplemental. What is the spatial resolution of the remote sensing data or the DEMs used to calculate geomorphons? Why were the specific spectral indices selected and from what satellite products were they computed?

[Response] We apologize for the oversight. Table 1 has been added, and the sources and spatial resolution of covariates are now listed. We have verified that none of these data have reported data quality issues in Central Europe.

Table 1. The predictor covariates used in the random forest model for inter-tree variations in wood density. The original 8-daily values of NDVI and NDWI were aggregated into a median (P50) and a standard deviation (STD) for the entire period.

| Variables | Description | Unit | Original resolution | Source |
|-----------|-------------|------|---------------------|--------|
| SNDPPT | Weight percentage of the sand particles (0.05–2 mm) | % | 250 m | SoilGrids database |
| NDWI | 8-daily Enhanced Vegetation Index (EVI) generated using the | 1 | 0.083º | MOD13A2 |

| | gridded daily surface reflectance product. | | | |
|---|---|---|---|---|
| NDVI | 8-daily Normalized Difference Vegetation Index (NDVI) generated using the gridded daily surface reflectance product. | 1 | | |
| Geomorphons | a pattern recognition approach to classification and mapping of landforms from digital elevation models (DEMs) | - | 30m | Jasiewicz & Stepinski (2013) |

3. A better description of the spatial sampling design is needed. The authors explain different height and age classes, and the total number of samples/trees/plots in Fig 1, and that 30+ samples were collected per tree, but the number of trees in each plot is not provided. This is quite important as it defines the data's hierarchical structure and should drive the analytical approach.

[Response] Thank you so much for your comments. We have added the sentence in the revised manuscript: "Our dataset includes the density of more than 48,000 samples taken from 2,920 trees, and from 391 forest plots in Poland, all carried out in the year 2018 (Figure 1). The number of trees per plot varies, averaging $6.7 \pm 3.0$ trees." (Page: 3; Lines: 133-134). Following your suggestion, we have performed a generalization linear mixed effects model to assess these data. Please refer to our response to comment #4 for further details.

4. With hierarchical data such as this, ANOVA is not an appropriate analytical approach. These data should be analyzed with a generalized linear mixed effects model so that the appropriate variance can be partitioned to the random effects (e.g. sample, tree, plot) rather than being fully attributed to the fixed effects (leaf type, family, species, age, dbh, etc). Without a proper hierarchical analysis, the results are challenging to interpret with confidence. Using a GLMM would also allow for all density samples to be pooled into a single analysis (with random effects for sample nested in tree nested in site), implicitly accounting for within tree variation in density, rather than averaging all density measurements per tree and then analyzing at the tree-level. This approach also increases degrees of freedom substantially, thereby increasing interpretive power.

**[Response]** Following the reviewer's suggestion, we applied a generalized linear model to the sample-level measurements (see Table R1), as well as two generalized linear mixed-effects models with "tree" and "plot" as random effects, respectively (see Table R2). The results indicate that while there is a relationship between wood density and variables such as tree species, families, leaf types, and ages, these factors alone account for only 17% of all the variation in wood density samples. This is consistent with our finding that the intra-tree variability of wood density within individual trees is substantial and cannot be fully explained by species, families, leaf types, or ages. However, when the data are grouped by tree, the model explains 74% of the variation in wood density (Model 2 in Table R2), using all sample values as predicted variable, and "tree" as a random factor. This result confirmed that our approach of linking tree-averaged wood density with tree species, families, leaf types, and ages is feasible. We acknowledge that calculating the tree-level average leads to the loss of intra-tree variation in wood density, which indeed cannot be explained by these factors.

**Table R1**. The result of generalized linear model for predicting wood density variation among all samples using fixed effects of species, families, leaf type, and ages.

| | Model 1 (Sample level) | | |
|---|---|---|---|
| | Beta | SE | p-value |
| Species | 0.002 | 0.001 | 0.059* |
| Families | -0.011 | 0.002 | <0.001** |
| Leaf types | 0.082 | 0.002 | <0.001** |
| Ages | 0.001 | <0.001 | 0** |
| | $R^2 = 0.17$ | | |

**Table R2**. The result of generalized linear mixed effects model for predicting wood density variation using fixed effects of species, families, leaf type, ages, and random effect of tree and plot.

| Model 2 (Tree level) | | | | Model 3 (Plot level) | | | |
|---|---|---|---|---|---|---|---|
| Fixed effects | Beta | SE | p-value | Fixed effects | Beta | SE | p-value |
| Species | -0.0001 | 0.004 | 0.98 | Species | -0.003 | 0.001 | 0.001** |
| Families | -0.013 | 0.007 | 0.08* | Families | 0.009 | 0.002 | 0** |
| Leaf types | 0.111 | 0.009 | 0** | Leaf types | 0.106 | 0.003 | 0** |

| Ages | 0.001 | <0.001 | 0** | Ages | 0.0007 | <0.001 | 0** |
|---|---|---|---|---|---|---|---|
| Random effect | | Variance | SD | Random effect | | Variance | SD |
| Intercept: Tree | | 0.010 | 0.099 | Intercept: Tree | | 0.06 | 0.079 |
| $R^2 = 0.74$ | | | | $R^2 = 0.43$ | | | |

We have added the following text to clarify this in the revised manuscript: "While our approach of linking tree-averaged wood density with tree species, families, leaf types, and ages is validated by the strong explanatory power of the generalized linear mixed-effects model (Table S2), we recognize that this method inherently averages out significant intra-tree variability. This variability, which cannot be fully accounted for by these factors alone, is an important aspect of wood density dynamics that warrants further investigation. Therefore, our findings should be interpreted with the understanding that the tree-level averages, while useful, may not capture the full complexity of wood density variations within individual trees." (Page: 9; Lines: 345-350)

5. It's not clear in the objectives why remote sensing data were used for the random forest modeling when they weren't used for any of the other analyses. Was this an attempt to be able to predict wood density in areas where samples were not collected but remote sensing data is available? If so, the authors need to explicitly state this objective.

[Response] Thanks for your comments. Actually, predicting wood density in areas without samples is not the purpose of this analysis. We have already achieve this globally in another analysis (https://onlinelibrary.wiley.com/doi/full/10.1111/gcb.17224). Here, our aim is to identify the key climatic, edaphic, and biotic factors controlling the spatial variations in observed wood density among trees, and to explore how wood density varies in response to these key factors. Since the climatic, edaphic, and vegetation properties (beyond wood traits) were not recorded during sample collection, we used the high-resolution satellite products and observation-based climate products to provide the local environmental conditions and vegetation state. In addition, we have added the sentence to state this clearly, as follow: "To investigate the key climatic, edaphic or vegetation-related factors influencing the spatial distribution of tree-level wood density, we extracted the relevant predicted variables from high-resolution satellite products and observation-based climate products, based on the latitude and longitude of samples, and employed a feature selection method (Jung and Zscheischler 2013) to identify the most significant predictors." (Page: 6; Lines: 277-280)

6. I don't understand why a feature selection procedure was used prior to random forest modeling (lines 130-131). Random forest has a built in feature selection process and is, for the most part, robust to a large number of predictors (assuming hyperparameters are appropriately defined). This two-part process may be omitting predictors that RF may have otherwise identified as important.

**[Response]** Thanks for your comment. We performed a feature selection prior to random forest modelling for following reasons:

(a) The total of the extracted climatic, edaphic or vegetation-related covariates variables from high-resolution products exceed 2,000, and computational limits prohibit the use of all covariates in the random forest modelling;

(b) In data-driven modelling, choosing the right predictor based on prior experience or knowledge can bring some biases. Therefore, we aimed to identify the most informative predictors directly for building the random forest model;

(c) The feature selection method used in this analysis can return multiple optimal solutions if they exist, which is important for interpreting the results (Jung and Zscheischler, 2013).

7. More information is needed on the random forest modeling methods, which package/library was used. How were hyperparameters selected or tuned? Were data centered or scaled in any way? Are random forests run at the sample-, tree-, or plot-level? If at tree- or plot-level how were density measurements aggregated? If averaged, were the individual density measurements normally distributed or was a median reported?

**[Response]** Thank you for pointing it out. We have added the information of the random forest modelling method, as follow: "The R package 'randomForest' was used in this analysis to build a random forest model with 500 trees. The tree-level wood density, calculated as the average value of all wood density samples within each tree, was randomly partitioned into training and testing subsets, with 80% of the measurements allocated to the training set and the remaining 20% reserved for testing." (Page: 6; Lines: 283-287)

Additionally, As the reviewer mentioned, we calculated the average of sample-level wood density to represent the tree-level wood density. Firstly, to determine if the average value is a good representation of population, we compared the mean and median values within each individual trees. As shown in Figure R2a, the difference between mean and median values are

minor (slope: 1.02, $R^2 = 0.99$). This suggests that the wood density samples within individual tree tend to follow a normal distribution, and the tree-level wood density can be effectively represented by the average of the sample-level values. Secondly, we examined whether the intra-tree variability of wood density differs among tree species. Figure R2b shows that, despite varying tree diameters, the magnitude of intra-tree variability is similar across tree species, with no significant differences observed.

[Figure]

**Figure R1**. (a) Comparison of the mean and median values of wood density within the individual trees. (b) The standard deviation (STD) of the sample-level wood density within the individual trees. Boxplots indicate the median, mean, minimum, maximum, and the 25th and 75th percentiles of wood density for eight tree species.

We also added the following text: "Tree-level wood density was calculated by the average of all samples within each individual trees. This method is used because there was no significant difference between the mean and median values of the samples (Figure S1a), indicating that wood density within an individual tree typically follows a normal distribution. Furthermore, the magnitude of intra-tree variability is consistent across eight tree species (Figure S1b)." (Page: 6; Lines: 274-277)

8. Out of bag estimates from random forest models are not "predictions," (e.g. lines 193, 265, 324) they report a variance explained only for the model fit to the data i.e. OOB estimates are for variance explained on data subsets withheld when each tree is defined. If the authors would like to report model "predictions" then the model needs to be tuned to a subset (70-80%) of the data (preferably with entire sites/plots omitted) and tested on another subset that has been entirely withheld from model tuning process. This also means that the 91% reported accuracy is likely overestimated in terms of predicting

wood density estimates from "new" or withheld data. In simpler language: as reported, the random forest model explains how well the model captures variation in the data, but that has little bearing on the ability of the model to predict (i.e. extrapolate) density estimates using spectral data from trees/areas not included in the dataset used to tune the model.

[Response] Thank you for your comments. Firstly, we completely agree with the reviewer on the importance of assessing the performance of model in predicting data not used during training. To assess this, we indeed partitioned the wood density data into training and testing subsets, randomly allocating 80% of measurements for training and the remaining 20% for testing. We have clarified this in the revised manuscript, and we provided the scatter plots comparing predictions and observations for both training and testing subsets (Figure R2). Secondly, the out of bag (OOB) error serves as a cross-validation accuracy metric. It helps avoid overfitting and assesses the prediction performance of the random forest model for samples not seen during the training. Therefore, we believe that the OOB estimates are also important and should be shown in the manuscript.

[Figure]

**Figure R2**. Comparison between predictions from the random forest model and observations for both training (blue dots) and testing (red dots) subsets.

9. Technically, spatial data inherently violate model assumptions of data independence for random forests. While it is increasingly common to use RF for modeling spatial data, the authors need to acknowledge potential biases in model performance and partial

effects of predictors due to spatial autocorrelation. There are several packages available to run random forests on spatial data such as the 'spatialRF' package in R.

**[Response]** Thanks for your comments. We agree with the reviewer that spatial autocorrelation can lead to a positively overestimation of the prediction skill of machine learning model (Ploton et al., 2020). Thus, a spatial blocked cross validation is indeed important for models used for prediction (as what we did in Yang et al. 2024).

However, in this analysis, the random forest model was designed to find the influencing factors for spatial variations in observed wood density, rather than to predict wood density. Thus, the risk of extrapolation is not a primary concern in this context. In this case, we extracted predictor variables from very high-resolution satellite or observation-based products. If spatial autocorrelation exists in these climatic, edaphic and vegetation variables, and if similar spatial autocorrelation is observed in the wood density measurement, it can help identify key factors influencing the spatial pattern of wood density.

**[Minor comments]** Line 17 – Typo: "representing" should be "represented"
**[Response]** Corrected.

Line 19 – Define or describe geomorphons at first mention rather than in following sentences.
**[Response]** We have rewritten this part as follows: "Geomorphons, which are landform elements derived from digital elevation models (DEM) and soil sand context provided insights into wood density variations. Lower wood density values were linked to landforms characterized by low geomorphons, such as summit, ridge, or shoulder. Conversely, higher wood density was found in landforms with high geomorphons, including valley, depression, or hollow areas." (Page: 1; Lines: 19-22)

Lines 64 and 76 – This study looks at variation \*among\* trees (i.e. comparison among many) rather than between trees – "between" implies a comparison between two individuals.
**[Response]** Corrected

Line 80 – What does it mean that trees were aged 5 years?
**[Response]** In this study, we used the tree samples from trees older than five years. This is because the younger trees were not selected during the sampling process, as they would need

to be cut down for inclusion in the study. We have clarified this in the revised manuscript: "Since the sampling process involves cut down trees, only those older than 5 years were included in this study. These trees, representing a range of species, had their relevant information such as latitude, longitude, age, and species type recorded." (Page: 3; Lines: 134-136)

**References**

Jung, M., & Zscheischler, J. (2013). A guided hybrid genetic algorithm for feature selection with expensive cost functions. Procedia Computer Science, 18, 2337-2346.

Ploton, P., Mortier, F., Réjou-Méchain, M., Barbier, N., Picard, N., Rossi, V., ... & Pélissier, R. (2020). Spatial validation reveals poor predictive performance of large-scale ecological mapping models. Nature Communications, 11(1), 4540.

Yang, H., Wang, S., Son, R., Lee, H., Benson, V., Zhang, W., ... & Carvalhais, N. (2024). Global patterns of tree wood density. Global Change Biology, 30(3), e17224.

---

## Author Comment (AC4)

[revised manuscript text omitted]

Large variability in wood density between trees has been reported. Firstly, wood density varies considerably across different tree species, genera, or families. Thurner et al. (2014) assessed the wood density measurements from Global Wood Density Database (Chave et al. 2006; Zanne et al. 2009), and found that, on average, broadleaf trees have higher wood density than needleleaf trees, but even within the same genus, significant divergence in wood density can be observed. Furthermore, the variation in wood density is closely linked to tree growth conditions, which encompass factors such as climate, nutrient availability, and soil characteristics. For example, previous regional studies have reported that wood density tends to increase with higher growth temperature (Thomas et al. 2005; Sweson and Enquist 2007) and lower elevation (Sungpalee et al. 2009), and the soil water availability, which has a negative impact on wood density in wet biomes but a positive impact in dry biomes (Rocha et al. 2020). Moreover, at the microscopic level, wood density is influenced by the characteristics of tracheid cells. For example, Gryc et al. (2011) found that in coniferous trees, thin cell walls resulting from fast growth due to lower competition for light and space are typically associated with low wood density. Conversely, thick cell walls, which result from slower growth, are related to high wood density. Additionally, the growth rate of individual trees can vary over their lifespan, leading to variations in wood density. Generally, young trees grow quickly while mature trees grow steadily (Bowman et al., 2012). Thus, the growth strategy between trees and the development stage of an individual tree's lifespan can play a role in shaping wood density gradients.

In this study, we use a novel dataset of wood density measurements collected from forests in Poland to investigate both inter-tree and intra-tree variations in wood density. The primary objectives of this study are as follows:

1. Determine the magnitude of inter-tree variations in wood density. We aim to explore how factors such as leaf type, tree family, tree species, and location contribute to the observed inter-tree variations. Additionally, we seek to understand how biotic and abiotic factors influence wood density variations among trees.

2. Examine how wood density changes with tree height (vertical density profiles), radius (radial density profiles), and different directions (northern or southern parts of discs) within individual trees. We aim to explore the underlying reasons behind these different vertical and radial density profiles within individual trees.

3. Compare the extent of inter-tree and intra-tree variations in wood density for the tree species or forest plots analyzed. We aim to determine which variation is larger and provide recommendations for estimating wood density at a large-scale.

Many previous studies have assessed the relationship between environmental conditions and wood density using data from limited the mature forest plots (Baker et al 2004; Dias et al 2018; Phillips et al 2019). The dataset used in this work is unique for Central Europe and, although it was only collected in Poland, covers age, habitat and height distributions characteristic of this part of the world. By addressing these research objectives, we aim to enhance our understanding of wood density variations both across and within trees, and provide insights into the estimation of wood density on a broader scale.

**2 Methods**

**2.1 Study site and wood density sample collection**

Our dataset includes the density of more than 48,000 samples taken from 2,920 trees, and from 391 forest plots in Poland, all carried out in the year 2018 (Figure 1). The number of trees per plot varies, averaging $6.7 \pm 3.0$ trees. Since the sampling process involves cut down trees, only those older than 5 years were included in this study. These trees, representing a range of species, had their relevant information such as latitude, longitude, age, and species type recorded. The dataset consists of eight common tree species (belonging to three families): *Pinus sylvestris* (Pinaceae), *Picea abies* (Pinaceae), *Abies alba* (Pinaceae), *Larix decidua* (Pinaceae), *Quercus robur* (Fagaceae), *Fagus sylvatica* (Fagaceae), *Betula pendula* (Betulaceae), and *Alnus glutinosa* (Betulaceae). Moreover, specific divisions within the dataset exist for certain species. For example, the plots of *Pinus sylvestris* and *Quercus robur* are categorized into two groups based on soil fertility, denoted as "low fertile soils" and "fertile soils" respectively. Similarly, the plots of *Picea abies* and *Fagus sylvatica* were classified into two groups according to the elevation of the plots, labeled as "lowlands" and "highlands and mountains" respectively. Low plots are typically located in lowlands, ranging from 0 to 300 m asl. In this study, field plots were selected from elevations not exceeding 100 m asl. In contrast, high plots begin at 300 m asl, encompassing both uplands and mountainous areas. In Polish conditions, these high

plots extend up to 1600 m asl. These distinctions within the species-specific plots allow for a more detailed exploration of the impacts of environmental conditions on wood density.

95   Based on the leaf type and leaf habit, the eight species in our analysis belong to three plant function types (PFTs) categories: evergreen needleleaf forest (ENF), deciduous needleleaf forest (DNF) and deciduous broadleaf forest (DBF). One family, Pinaceae, has both evergreen needleleaf and deciduous needleleaf species. Taking into account both the families and PFTs, the eight species can be further divided into four types, i.e. Pinaceae_ENF, Pinaceae_DNF, Fagaceae_DBF and Betulaceae_DBF. The trees population were divided arbitrarily into nine age classes: The age of trees are 0-20 years, 20-40

100  years, 40-60 years, …, 140-160 and 160-180 years, respectively. Note that the age of the tree is determined by counting the rings on wood discs obtained from the bottom of a trunk. Furthermore, the trees were classified into seven height classes and six diameter-at-breast-height (DBH) classes: The heights classes were divided into following seven categories: <10 meters, 10-15 meters, 15-20 meters, 20-25 meters, 25-30 meters, 30-35 meters and >35 meters. The DBH of trees were divided into six categories: <100 centimeters, 100-200 centimeters, 200-300 centimeters, 300-400 centimeters, 400-500 centimeters, and

105  >500 centimeters.

[Figure]

Figure 1: (a) The number of samples (solid-colored bars), trees (light-colored bars), and forest plots (transparent bars) utilized for density measurements across the eight species. (b) Location of 391 forest plots in Poland (symbols as in panel a).

For the analysis of the intra-tree variation in wood density, a total of 1,886 trees were included, and for each tree, more than 30 wood samples were collected from tree cores obtained from discs using a sharp increment borer. These trees were specifically selected as they were dead but had not undergone wood drying. Each tree was divided into three equal parts, and a disc was obtained from the middle of each part. The three discs were labeled as "bottom," "middle," and "top" based on their respective positions within the tree. Each disc was cut from north to south to obtain a strip of wood. The samples were divided and numbered into two rays: north and south, starting from the core to the peripheral part. This method allowed for the estimation of variation in the radial density of wood. The number of samples obtained for each disc varied depending on the width of the disc, but each disc typically yielded more than 10 samples along these radial directions. Standardized wood density samples, measuring 2×2×3 cm, were cut from the strips, which were dried in a dryer at temperature of $103 \pm 2$ °C to an absolutely dry state. After the samples cooled down in the desiccator, the linear dimensions of the samples were measured using an electronic caliper, and their weight was measured on a laboratory scale. The stereometric density was then calculated from the classical mass/volume formula. During measurement, each density sample was examined for wood defects such as compression and tension wood, knots, resin wood, cracks, abnormal shapes after drying, and other irregularities. In this study, only defect-free samples were selected for further analysis, adhering to the standards for small sample density measurement.

**2.2 Study site and wood density sample collection**

To assess the inter-tree variations, we computed the mean wood density for each tree. Subsequently, we employed analysis of variance (ANOVA) to partition the overall variations in tree-level wood density (n = 2,920) across different levels including leaf habit, leaf type, family, species, and age classes. The total variance is calculated as:

$$\sum_{i=1}^{t} \sum_{j=1}^{n_i} (X_{ij} - \bar{X})^2 , \qquad (1)$$

where $X_{ij}$ is the $j$th wood density from class $i$, and there are $n_i$ wood density samples in class $i$, and $\bar{X}$ is the average of all the wood density samples. And variance explained by leaf habit/leaf type/family/species/age is calculated as:

$$\sum_{i=1}^{t} n_i (\overline{X_{i \cdot}} - \bar{X})^2 , \qquad (2)$$

where $\overline{X_{i \cdot}}$ is the average of wood density of class $i$.

To assess the intra-tree variations, we examined the differences in wood density among the bottom, middle and top parts of trees. In addition to ANOVA analysis, we conducted multiple comparison tests to evaluate the significance of wood density differences between any two of the bottom, middle, and top discs.

To account for the influence of location, we aggregated tree-level wood density measurements based on geographical proximity. Specifically, we considered two criteria: (1) trees located within a short distance of each other, and (2) trees falling within the same fine-resolution grid cell. Using the first criterion, trees were considered to be at the same location if the distance between them was less than 100 meters (or 500 meters). This resulted in the distribution of 2,920 trees across 382 unique

locations (or 372 unique locations). Using the second criterion, a pre-defined grid mesh with resolutions of 0.05° (or 0.1°) was utilized. Trees falling within the same grid cell were considered to be at the same location. To analyze the impact of location, we applied the same ANOVA methodology used previously to partition the total variations in tree-level wood density into different locations.

145

Tree-level wood density was calculated by the average of all samples within each individual trees. This method is used because there was no significant difference between the mean and median values of the samples (Figure S1a), indicating that wood density within an individual tree typically follows a normal distribution. Furthermore, the magnitude of intra-tree variability is consistent across eight tree species (Figure S1b). To investigate the key climatic, edaphic or vegetation-related factors influencing the spatial distribution of tree-level wood density, we extracted the relevant predicted variables from high-resolution satellite products and observation-based climate products, based on the latitude and longitude of samples, and employed a feature selection method (Jung and Zscheischler 2013) to identify the most significant predictors. Based on this selection, six important covariates were chosen, including vegetation indexes, vegetation water content, soil texture, and topographic characteristics (refer to Table 1). These selected covariates were then used to train a random forest model in the cross-validation analysis. The random forest model, based on decision tree ensembles, has been shown to have a better performance than the neural networks for handling tabular data (Grinszstajn *et al.*, 2022). The R package 'randomForest' was used in this analysis to build a random forest model with 500 trees. The tree-level wood density, calculated as the average value of all wood density samples within each tree, was randomly partitioned into training and testing subsets, with 80% of the measurements allocated to the training set and the remaining 20% reserved for testing. To evaluate the performance of the model, we assessed its efficiency using the Out-of-bag (OOB) $R^2$ metric, which yielded a value of 0.91. Additionally, in order to gain insights into how these selected covariates influence wood density within the random forest model, we computed the SHAP (Shapley Additive exPlanations) values for each covariate. These values represent the difference between the model's prediction and the null model (Lundberg and Lee, 2017). By examining the SHAP values, we can gain a better understanding of the individual contributions of each covariate to the prediction of wood density.

165

**Table 1. The predictor covariates used in the random forest model for inter-tree variations in wood density. The original 8-daily values of NDVI and NDWI were aggregated into a median (P50) and a standard deviation (STD) for the entire period.**

| Variables | Description | Unit | Original resolution | Source |
|-----------|-------------|------|---------------------|--------|
| SNDPPT | Weight percentage of the sand particles (0.05–2 mm) | % | 250 m | SoilGrids database |
| NDWI | 8-daily Enhanced Vegetation Index (EVI) generated using the gridded daily surface reflectance product. | 1 | 0.083° | MOD13A2 |
| NDVI | 8-daily Normalized Difference Vegetation Index (NDVI) generated using the gridded daily surface reflectance product. | 1 | | |
| Geomorphons | a pattern recognition approach to classification and mapping of landforms from digital elevation models (DEMs) | - | 30m | Jasiewicz & Stepinski (2013) |

**3 Results**

**3.1 Inter-tree variation in wood density**

Figure 2a shows the distribution of tree-level wood density for eight tree species, which were classified into three families, and three PFTs (also see Table S1). Overall, the variation in wood density among species is greater than the variation within each species. Consistent with the findings in Thurner et al. (2014), our results indicate that the mean wood density of evergreen needleleaf forests, that is *Pinus sylvestris*, *Picea abies*, *Abies alba* species, is lower than the density of deciduous needleleaf forests (*Larix decidua* species), and the mean wood density of needleleaf forests is lower than that of broadleaf forests, including *Quercus robur*, *Fagus sylvatica*, *Betula pendula*, and *Alnus glutinosa* species. When considering tree families, the mean wood density of Pinaceae is slightly lower than that of Betulaceae (Pinaceae: $0.44 \pm 0.06$ < Betulaceae: $0.53 \pm 0.06$; *p*-value < 0.001 according to the left-tailed *t*-test), and significantly lower than that of Fagaceae (Fagaceae: $0.67 \pm 0.04$; *p*-value = 0 according to the left-tailed *t*-test). These findings align with the general patterns observed in previous research, highlighting the differences in wood density among tree species and families.

[Figure]

**Figure 2: (a) Boxplots of tree-level wood density. On each box, the central bar indicates the median and the dot indicates the mean of wood density; the bottom and top edges indicate the 25th and 75th percentiles; the whiskers extend to all data points except**
185    **outliers (which are plotted individually as small red dots). Eight tree species belong to three families (Pinaceae, Fagaceae and Betulaceae), and were classified into two categories according to their leaf habit or types: evergreen (E) or deciduous (D) trees, needleleaf (N) or broadleaf (B) trees. Family, leaf habit and leaf type are labelled on the top of boxplot. (b) The fraction of variance of tree-level wood density explained by the leaf habit (two levels), leaf type (two levels), families (three levels), species (eight levels), age (nine levels), height (seven levels), DBH (six levels) and locations. The trees with geographic distance less than 100 or 500 m, or**
190    **in the same fine-resolution grid cell (i.e., 0.05, 0.1 degree) are considered as in the same location.**

A quantitative analysis indicates that species, families, leaf types, and leaf habits explain a massive portion of the variance in tree-level wood density, accounting for 85%, 80%, 63% and 58% respectively (Figure 2b). In contrast, tree location is not a discriminating factor, and it only explains less than 3% of the variance in tree-level wood density, regardless of the method related to distances or grid sizes used to identify trees within the same location (see Methods, section 2.2 Analysis of inter-tree

195    variation in wood density). The reason is that the geographical location does not differentiate the tree species in Poland, as different tree species with varying wood density are distributed across similar locations in Poland, as shown in Figure 1b. This variance in wood density, unexplained by location, could be related to local environmental conditions such as vegetation characteristics, soil properties and topography. While our approach of linking tree-averaged wood density with tree species,

families, leaf types, and ages is validated by the strong explanatory power of the generalized linear mixed-effects model (Table
S2), we recognize that this method inherently averages out significant intra-tree variability. This variability, which cannot be
fully accounted for by these factors alone, is an important aspect of wood density dynamics that warrants further investigation.
Therefore, our findings should be interpreted with the understanding that the tree-level averages, while useful, may not capture
the full complexity of wood density variations within individual trees.

[Figure]

**Figure 3: (a) Boxplots of tree-level wood density for needleleaf and broadleaf trees at nine different age classes. The higher classes, the older ages. (b) Boxplots of tree-level wood density for needleleaf and broadleaf trees with different height classes. The height of tree increases with the number of height class. (c) Boxplots of tree-level wood density for needleleaf and broadleaf trees with different DBH classes. The DBH of tree increases with the number of DBH class. (d) comparison of tree-level wood density for one specific species but growing in the low fertile or fertile soils, or growing at lowlands or highlands (mountains). Two asterisk indicates the significant difference in the mean of two samples (via t-test, 0.001 significance level).**

The contributions of tree age, height, and breast height diameter (DBH) to the variance in wood density among trees are
relatively low, accounting for 6%, 4%, and 2% respectively (Figure 2b). Note that height, DBH and age are interrelated and
not independent variables, and their collinearity may result in an overestimation of the contribution of each individual factor.
When analyzing needleleaf and broadleaf trees separately, their effects become more apparent (Figure 3a-c). First, regarding
the impacts of tree ages (Figure 3a), for broadleaf trees, wood density tends to increase with tree age up to approximately 140
years (class VII), after which it stabilizes. For needleleaf trees, wood density also exhibits an increase in wood density with
age up to 140 years (class VII). Since there are no observation of wood density from older needleleaf trees, it remains unclear
whether wood density would continue increase or stabilized beyond 140 years. Second, both needleleaf and broadleaf trees
show an increase in mean wood density with height and DBH classes, especially for tall broadleaf trees (height class ≥ 4, i.e.,
tree height ≥ 20m). However, the variance in wood density within these height and DBH classes are large, resulting in no

[revised manuscript text omitted]

Baker, T. R., Phillips, O. L., Malhi, Y., Almeida, S., Arroyo, L., Di Fiore, A., ... & Vasquez Martinez, R. 2004. Variation in wood density determines spatial patterns inAmazonian forest biomass. Global Change Biol 10(5), 545-562.

395   Barnett, J. R., & Jeronimidis, G. (Eds.). 2003. Wood Quality and its Biological Basis. CRC Press.

Brando, P.M., Nepstad, D.C., Balch, J.K., Bolker, B., Christman, M.C., Coe, M., Putz, F.E., 2012. Fire-induced tree mortality in a neotropical forest: the roles of bark traits, tree size, wood density and fire behavior. Global Change Biol 18, 630-641.

Chao, K.J., Phillips, O.L., Gloor, E., Monteagudo, A., Torres-Lezama, A., Martínez, R.V., 2008. Growth and wood density
400        predict tree mortality in Amazon forests. J Ecol 96, 281-292.

Chave, J., Andalo, C., Brown, S., Cairns, M.A., Chambers, J.Q., Eamus, D., ... , Yamakura, T., 2005. Tree allometry and improved estimation of carbon stocks and balance in tropical forests. Oecologia 145, 87-99.

Chave, J., Muller-Landau, H.C., Baker, T.R., Easdale, T.A., Steege, H.T., Webb, C.O., 2006. Regional and phylogenetic variation of wood density across 2456 neotropical tree species. Ecol Appl 16, 2356-2367.

405   Chave, J., Coomes, D., Jansen, S., Lewis, S.L., Swenson, N.G., Zanne, A.E., 2009. Towards a worldwide wood economics spectrum. Ecol Lett 12, 351-366.

Dias, A., Gaspar, M. J., Carvalho, A., Pires, J., Lima-Brito, J., Silva, M. E., & Louzada, J. L. 2018. Within-and between-tree variation of wood density components in Pinus nigra at six sites in Portugal. Ann For Sci 75, 1-19.

Gao, B. C. 1996. NDWI—A normalized difference water index for remote sensing of vegetation liquid water from space.
410        Remote Sens Environ 58(3), 257-266.

Grinsztajn, L., Oyallon, E., & Varoquaux, G. 2022. Why do tree-based models still outperform deep learning on typical tabular data?. Adv Neural Inf Process Syst 35, 507-520.

Gryc, V., Vavrčík, H., & Horn, K, 2011. Density of juvenile and mature wood of selected coniferous species. J For Sci 57(3), 123-130.

415   Huete, A., Justice, C., & Van Leeuwen, W. 1999. MODIS vegetation index (MOD13). Algorithm theoretical basis document, 3(213), 295-309.

Jasiewicz, J., & Stepinski, T. F. 2013. Geomorphons—a pattern recognition approach to classification and mapping of landforms. Geomorphology, 182, 147-156.

Jung, M., Zscheischler, J., 2013. A guided hybrid genetic algorithm for feature selection with expensive cost functions.
420        Procedia Comput Sci 18, 2337-2346.

King, D.A., Davies, S.J., Supardi, M.N., Tan, S., 2005. Tree growth is related to light interception and wood density in two mixed dipterocarp forests of Malaysia. Funct. Ecol. 19, 445-453.

Kraft, N.J., Metz, M.R., Condit, R.S., Chave, J., 2010. The relationship between wood density and mortality in a global tropical forest data set. New Phytol 188, 1124-1136.

425 Liang, X., Ye, Q., Liu, H., Brodribb, T.J., 2021. Wood density predicts mortality threshold for diverse trees. New Phytol 229, 3053-3057.

Lundberg, S.M., Lee, S.I., 2017. A unified approach to interpreting model predictions. Advances in neural information processing systems, 30.

Pallardy, S. G. 2008. Physiology of Woody Plants. Academic press.

430 Phillips, O. L., Sullivan, M. J., Baker, T. R., Monteagudo Mendoza, A., Vargas, P. N., & Vásquez, R. 2019. Species matter: wood density influences tropical forest biomass at multiple scales. Surv Geophys 40, 913-935.

Rocha, S.M.G., Vidaurre, G.B., Pezzopane, J.E.M., Almeida, M.N.F., Carneiro, R.L., Campoe, O.C., ... & Figura, M.A., 2020. Influence of climatic variations on production, biomass and density of wood in eucalyptus clones of different species. For Ecol Manage 473, 118290.

435 Santoro, M., Cartus, O., Mermoz, S., Bouvet, A., Le Toan, T., Carvalhais, N., ... , Seifert, F.M., 2018. A detailed portrait of the forest aboveground biomass pool for the year 2010 obtained from multiple remote sensing observations. In EGU general assembly conference abstracts: 18932.

Swenson, N.G., Enquist, B.J., 2007. Ecological and evolutionary determinants of a key plant functional trait: wood density and its community-wide variation across latitude and elevation. Am J Bot 94, 451-459.

440 Sungpalee, W., Itoh, A., Kanzaki, M., Sri-ngernyuang, K., Noguchi, H., Mizuno, T., Sorn-ngai, A., 2009. Intra-and interspecific variation in wood density and fine-scale spatial distribution of stand-level wood density in a northern Thai tropical montane forest. J Trop Ecology, 25, 359-370.

Thomas, D.S., Montagu, K.D., Conroy, J.P., 2007. Temperature effects on wood anatomy, wood density, photosynthesis and biomass partitioning of Eucalyptus grandis seedlings. Tree Physiol 27, 251-260.

445 Thurner, M., Beer, C., Santoro, M., Carvalhais, N., Wutzler, T., Schepaschenko, D., ..., Schmullius, C., 2014. Carbon stock and density of northern boreal and temperate forests. Global Ecol Biogeogr 23, 297-310.

Wiemann, M.C., Williamson, G.B., 2002. Geographic variation in wood specific gravity: effects of latitude, temperature, and precipitation. Wood Fiber Sci 34, 96-107.

Woodcock, D., Shier, A., 2002. Wood specific gravity and its radial variations: the many ways to make a tree. Trees, 16, 437-
450 443.

Zanne, A.E., Westoby, M., Falster, D.S., Ackerly, D.D., Loarie, S.R., Arnold, S.E., Coomes, D.A., (2010). Angiosperm wood structure: global patterns in vessel anatomy and their relation to wood density and potential conductivity. Am J Bot 97, 207-215.